EMBO
reports

# Human RAP1 specifically protects telomeres of senescent cells from DNA damage

Liudmyla Lototska[1,2,†], Jia-Xing Yue[2,‡], Jing Li[2,‡], Marie-Josèphe Giraud-Panis[2], Zhou Songyang[3,4,5], Nicola J Royle[6] iD, Gianni Liti[2], Jing Ye[1,*] iD, Eric Gilson[1,2,7,**] iD & Aaron Mendez-Bermudez[1,2,#,***] iD

## Abstract

Repressor/activator protein 1 (RAP1) is a highly evolutionarily conserved protein found at telomeres. Although yeast Rap1 is a key telomere capping protein preventing non-homologous end joining (NHEJ) and consequently telomere fusions, its role at mammalian telomeres *in vivo* is still controversial. Here, we demonstrate that RAP1 is required to protect telomeres in replicative senescent human cells. Downregulation of RAP1 in these cells, but not in young or dividing pre-senescent cells, leads to telomere uncapping and fusions. The anti-fusion effect of RAP1 was further explored in a HeLa cell line where RAP1 expression was depleted through an inducible CRISPR/Cas9 strategy. Depletion of RAP1 in these cells gives rise to telomere fusions only when telomerase is inhibited. We further show that the fusions triggered by RAP1 loss are dependent upon DNA ligase IV. We conclude that human RAP1 is specifically involved in protecting critically short telomeres. This has important implications for the functions of telomeres in senescent cells.

**Keywords** chromosome fusions; non-homologous end joining; RAP1; replicative senescence; telomeres

**Subject Categories** Autophagy & Cell Death; DNA Replication, Recombination & Repair

## Introduction

Mammalian telomeres are composed of tandem arrays of the TTAGGG repeat sequence ending with a G-rich single-strand overhang. Over the course of evolution, mammalian telomeres invented the shelterin complex, a very elegant way to protect chromosome extremities from various DNA damaging insults, including telomere fusion events. In humans, shelterin is composed of six specialized proteins. TRF1 and TRF2 proteins bind the double-stranded part of the telomeres, while POT1 binds the single-stranded portion and interacts with TPP1 through TIN2. Finally, RAP1, one of the most evolutionarily conserved shelterin proteins, binds to telomeric DNA via its interaction with TRF2 [1,2].

In budding yeast, Rap1 protects the chromosome ends against classical non-homologous end joining (c-NHEJ) [3]. It can do so either through its C-terminal RCT domain directly or via recruitment of two other proteins, Sir4 and Rif1 [4]. Although yeast Rap1 is a key anti-fusion protein, conflicting results regarding its role as an anti-fusion factor in mammals have been reported. Indeed, mouse and human telomeres lacking RAP1 did not develop DNA damage response activation [5,6] but were prone to recombination by homology-directed repair [5]. As an outcome, this can trigger telomere resection and fusions [7]. In this regard, Rai and co-workers identified that the N-terminal BRCT and the central MYB domains of RAP1 are important to prevent telomere-free fusions and signal-free ends. They showed that RAP1 in cooperation with its interacting partner TRF2 is required to fully repress PARP1 and SLX4 localization at telomeres and further t-loop resolution and telomere loss due to circle-mediated excision [7].

*In vitro*, human RAP1 has been shown to protect against NHEJ in cooperation with TRF2 [8–10]. Furthermore, it was demonstrated *in vivo* that artificially targeting RAP1 to telomeres in a TRF2-independent manner can mediate NHEJ [9]. Recently, Martinez and colleagues [11] have shown in mice that upon telomerase dysfunction, RAP1-deficient mice suffer from telomere end-to-end fusions and telomere loss. An emerging view is that RAP1 acts as a backup

1 Shanghai Ruijin Hospital, Shanghai Ruijin Hospital North, Shanghai Jiao Tong University School of Medicine, Université Côte d'Azur, CNRS, Inserm, International Research Laboratory in Hematology, Cancer and Aging, State Key Laboratory of Medical Genomics, Shanghai, China
2 Université Côte d'Azur, CNRS, INSERM, IRCAN, Medical School of Nice, Nice, France
3 Verna and Marrs McLean Department of Biochemistry and Molecular Biology, Baylor College of Medicine, Houston, TX, USA
4 School of Life Sciences, Sun Yat-sen University, Guangzhou, China
5 Zhongshan Ophthalmic Center, Sun Yat-sen University, Guangzhou, China
6 Department of Genetics, University of Leicester, Leicester, UK
7 Department of Genetics, CHU, Nice, France
*Corresponding author. Tel: +86 021 64370045 611110; E-mail: yj11254@rjh.com.cn
**Corresponding author. Tel: +33 4 93 37 77 82; E-mail: eric.gilson@unice.fr
***Corresponding author. Tel: +33 (0)4 93 37 70 17; E-mail: amendez@unice.fr
†Present address: Institute of Molecular Biology gGmbH (IMB), Mainz, Germany
‡Present address: State Key Laboratory of Oncology in South China, Collaborative Innovation Center for Cancer Medicine, Sun Yat-sen University Cancer Center, Guangzhou, China
#Lead author

anti-fusion factor in mammalian cells when its interacting partner TRF2 is dysfunctional but still bound to telomeres [12].

In this study, we present strong evidence that human RAP1 protects critically short telomeres. RAP1 prevents the accumulation of 53BP1 (p53-binding protein 1) at telomeres as well as protects them from fusions in senescent cells but not in cells that are actively dividing. In addition, we show that HeLa cells require RAP1 to protect their telomeres only upon telomerase inhibition, when telomeres get shorter. We further show that telomere fusion events are generated through the classical NHEJ pathway.

## Results and Discussion

### RAP1 is required for telomere protection in senescent fibroblasts

We passaged human primary lung fibroblasts (MRC-5) at 5% oxygen and we performed Western blotting at different population doublings (PDs) to determine the levels of RAP1 and TRF2 expression in young (PD 26), pre-senescent (PD 66), and senescent cells (PD 72) (Fig 1A). We defined a cellular culture as senescent when the number of cells did not increase for at least 4 weeks, SA-β-galactosidase (SA-β-Gal) activity was detected, and EdU incorporation accounted for < 1% of cells. As expected from replicative senescent cells and aging animal models, the levels of TRF2 greatly decreased (around 80%) in senescent cells [13–15]. However, the levels of RAP1 barely changed with a mild decrease of only 15% as seen previously [16,17].

In order to investigate the presence of TRF2 and RAP1 specifically at telomeres of senescent fibroblasts, we performed chromatin immunoprecipitation (ChIP) in MRC-5 cells with either anti-RAP1 or anti-TRF2 antibodies. The immunoprecipitate (IP) product was spotted on nylon membranes, and it was hybridized with a radioactively labeled telomere probe. We normalized the signal of the IP to that of the input to take into account the loss of telomere repeats during replicative senescence and thus to estimate the density of proteins bound along telomeric DNA. The density at telomeres of both TRF2 and RAP1 was reduced in senescent cells (PD 72) as compared to young cells (PD 30), but it was still detectable (Fig 1B). Interestingly, the density of telomere-bound RAP1 is reduced by only twofold in senescent cells compared to a reduction of more than fourfold for TRF2. This apparent discrepancy could stem from differences in the binding of RAP1 to TRF2 across the telomeric array.

It is possible that RAP1 binds preferentially the molecules of TRF2 found at the end of the telomeric array.

Next, we asked whether RAP1 is required for telomere protection during replicative senescence. To do so, we used an shRNA against RAP1 in MRC-5 cells of different population doublings (Fig EV1A) and we monitored telomere protection by analyzing the co-localization between 53BP1, a DNA damage response (DDR) protein, and a telomeric PNA probe (telomere dysfunction-induced foci, TIF assay) (Fig 1C). The frequency of TIFs increases with accumulating cell divisions as previously reported [18–21]. In agreement with previous studies failing to detect telomere dysfunction upon RAP1 disruption [5,6,22], RAP1 downregulation in young and pre-senescent cells did not increase the amount of TIFs. However, inhibition of RAP1 in senescent cells significantly increased TIF levels, showing that RAP1 protects telomeres against DNA damage checkpoint activation in senescent cells (Fig 1C). This increase was observed either by using lentivirus expressing an shRNA or by transient transfection of siRNAs. Notably, telomere damage was rescued by ectopic expression of RAP1, ruling out an off-target effect in our experimental settings.

Then, we measured total levels of damage by counting the number of 53BP1 spots in the cells and, as expected, total damage increased progressively as the cells approach senescence; however, RAP1 downregulation did not change the overall levels of damage, indicating the effect of RAP1 is telomere-specific (Fig 1C).

Importantly, RAP1 loss did not induce mean telomere length shortening, as revealed by Southern blotting, nor critically short telomeres, as revealed by the PCR-based STELA method (Fig EV1B and C). Thus, the increase of TIFs is unlikely to be the consequence of an excess of telomere shortening in RAP1-compromised cells. Together, these results show that the protective role of RAP1 is specific of senescent cells, as compared to dividing (young or pre-senescent) cells.

### RAP1 prevents telomere fusions in senescent fibroblasts

Next, we investigated whether RAP1 protects telomeres of senescent cells from end-to-end fusions by means of a PCR-based method that relies on the use of subtelomeric DNA primers to amplify fusions between different chromosome ends (Fig EV2A) [23,24]. We performed PCRs with a set of three different subtelomeric primers that bind to the ends of approximately 22 chromosomes. We found a very low number of telomere fusions in young and pre-senescent fibroblasts that remained nearly unchanged upon RAP1

**Figure 1. RAP1 protects telomeres against DNA damage.**

A Western blotting showing the expression of RAP1 and TRF2 in MRC-5 cells of different population doublings (PD). Senescent cells (PD 72) were left in culture for at least 4 weeks before harvesting for analysis.

B ChIP analysis of young (PD 30) and senescent (PD 72) MRC-5 cells performed with either anti-RAP1, anti-TRF2, or an IgG antibody. The immunoprecipitated and input products were loaded into a slot blot membrane and hybridized with a telomere probe, and stripped and hybridized to an Alu probe in order to determine unspecific binding. Quantification was performed by normalizing the telomere signal of the immunoprecipitated to that of the input. Data represent mean ± SD of three biological replicates.

C Immunofluorescence detection of 53BP1 (red) and a FISH probe staining telomeres (green) in young (PD 26), pre-senescent (PD 66), and senescent (PD 72 + 4 weeks) MRC-5 fibroblasts. The upper graph shows the percentage of telomere co-localizing with 53BP1 (TIFs). Total DNA damage was measured by immunofluorescence detection of 53BP1. The lower graph shows the total number of 53BP1 foci per nucleus. Approximately 40–50 cells were analyzed per replicate and per condition. Bars represent SEM of two biological replicates.

Data information: Statistical analyses were performed using Mann–Whitney U-test (*P < 0.05; **P < 0.001; ***P < 0.0001). Scale bar = 10 μm.
Source data are available online for this figure.

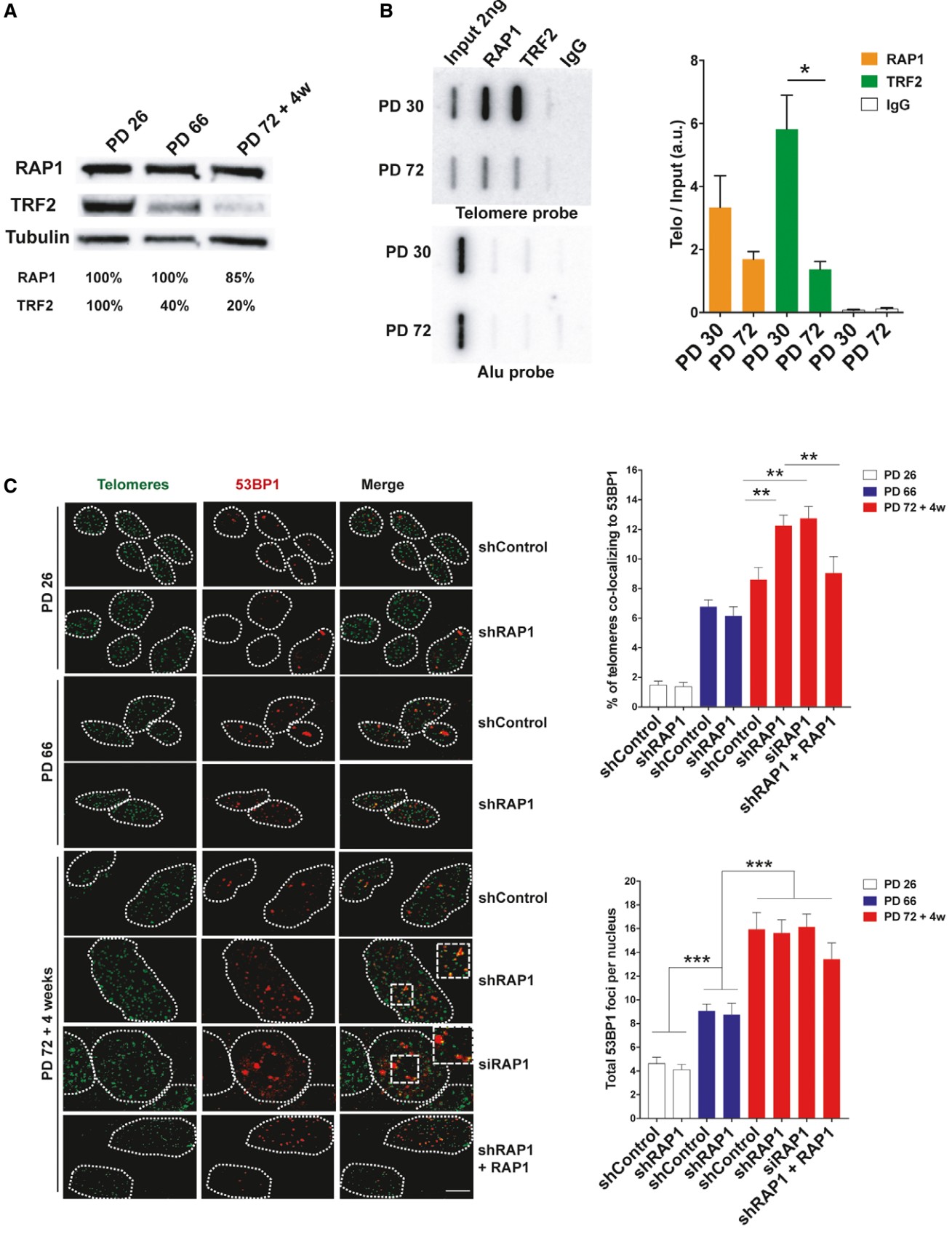

**Figure 1.**

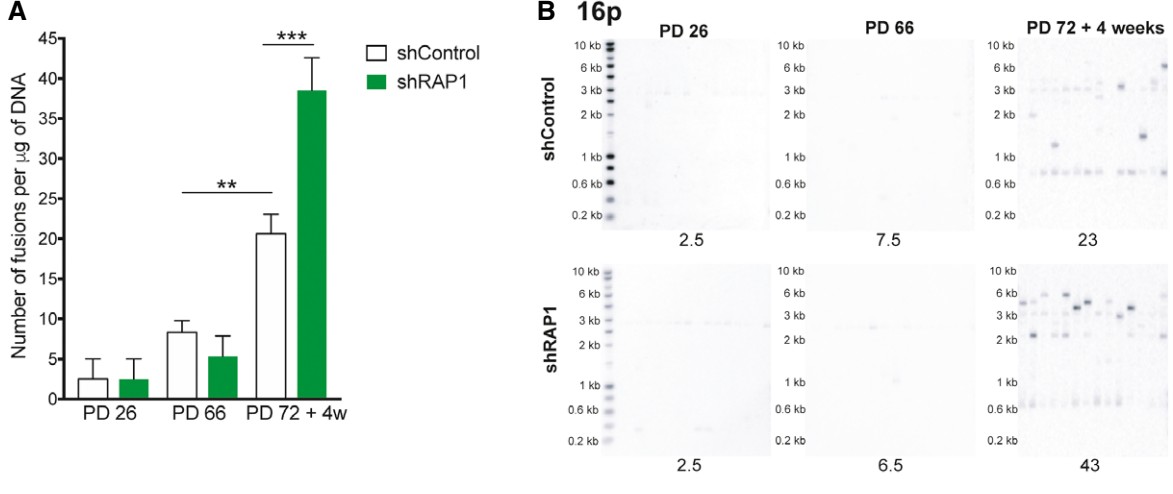

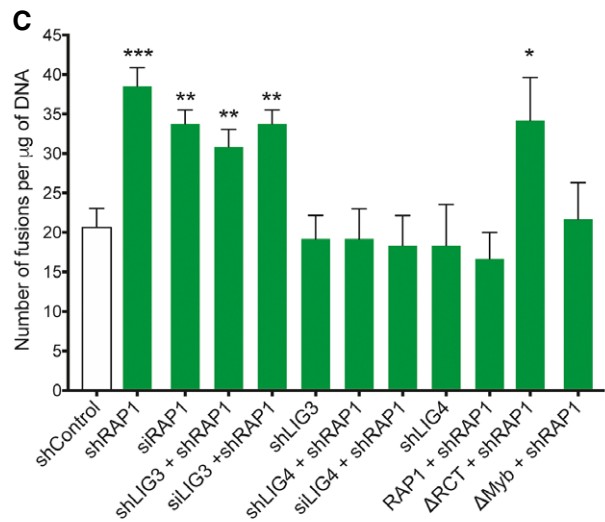

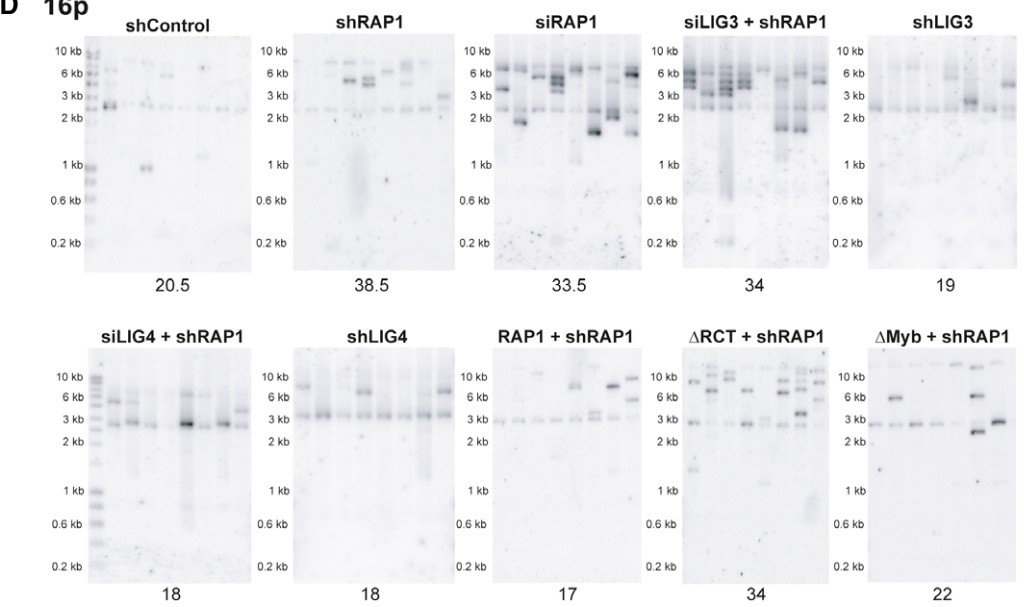

**Figure 2.**

**Figure 2. RAP1 prevents telomere fusions in senescent cells.**

A   Number of telomere fusions performed by PCR using 3 subtelomeric primers in the same reaction (16p1, 21q1, and XpYpM). The primer 16p1 is able to bind the subtelomeric region of 1p 9q, 12p, 15q, 16p, XqYq, and the interstitial 2q14 region; the 21q1 primer binds the subtelomeric region of 1q, 2q, 5q, 6q, 6p, 8p, 10q, 13q, 19p, 19q, 21q, 22q, and the interstitial 2q13 site. The PCR-fusion assay was performed in MRC-5 primary fibroblasts of different population doublings (PD) with or without depletion of RAP1 (shRAP1 or shControl, respectively). Lentiviral infection was performed for 10 days. Data represent mean $\pm$ SD of three biological replicates. Statistical analyses were performed using Mann–Whitney $U$-test (**$P < 0.001$; ***$P < 0.0001$).

B   Southern blotting showing the hybridization of the 16p probe for the conditions described in (A). Number of fusions per 1 µg of DNA is shown at the bottom of the gels.

C, D   Number of fusions per 1 µg of DNA in MRC-5 senescent cells. Lentiviral transfections with shRNAs were carried out for 10 days while transfections with siRNAs for 6 days (two subsequent transfections of 3 days each). Representative Southern blot membranes hybridized with the 16p probe are shown. Data represent mean $\pm$ SD of three biological replicates. Statistical analyses were performed using Mann–Whitney $U$-test (*$P < 0.05$; **$P < 0.001$; ***$P < 0.0001$).

downregulation. In contrast, we found a significant fusion frequency increase in senescent cells which increase to even higher levels when RAP1 was depleted (Figs 2A and B, and EV2B and C) showing that RAP1 prevents fusion events in particular of senescent cells.

Next, we asked whether the fusions were the result of the classical or alternative NHEJ pathway by depleting cells of either DNA ligase IV (*LIG4*) or DNA ligase III (*LIG3*) (Figs 2C and D, and EV2D) [25]. Only cells downregulated for LIG4 did not increase fusion frequencies upon RAP1 inhibition showing that the fusions triggered by RAP1 loss are dependent on the classical NHEJ pathway.

In order to dissect which RAP1 domain(s) protect(s) senescent fibroblasts against fusions, we generated mutants of RAP1 that were lacking either the TRF2-binding site of RAP1 (RCT) or the Myb domain of the protein and transduced them (or the full-length protein) together with a lentivirus expressing an shRAP1 sequence in senescent MRC-5 cells (Fig EV2E). The increase in chromosome fusions was rescued by infecting cells with the full-length RAP1 or the Myb-deficient mutant. However, expression of the RCT mutant did not rescue fusions, suggesting that the low levels of TRF2 found at telomeres of senescent cells are able to recruit RAP1 to protect telomeres against fusion events (Fig 2C and D).

**Telomere fusions occur between critically short telomeres**

The PCR-based method we used to detect fusion events relies on the use of three subtelomeric primers in the same PCR to increase the probability to detect fusions. We observed several PCR bands of the same size, more likely representing stable fusion events that suffer clonal expansion and thus we considered those ones as a single fusion.

To go deeper into the characterization of the fusion events, we pooled several PCRs of senescent cells transduced with shControl or shRAP1. We cleaned them up by phenol–chloroform and we performed long-range sequencing by Nanopore of 1 µg purified PCR products. After careful analysis of the sequence reads, we obtained 47 different fusion events in shControl condition compared to 84 in shRAP1 (Table EV1). Most of the fusions in the shControl condition lacked telomeric repeats at the junction of the fusion (Fig 3A and B), and even more, part of the subtelomeric DNA is resected as reported previously [23]. This explains

why most of the fusion events are shorter than the minimum predicted size with the primers used (e.g., 7.5 kb for fusion events relying on the 16p1 primer). Interestingly, we found that fusions generated in the shRAP1 condition contained more telomeric DNA repeats at the junction of the fusions with a mean length of 280 bp compared to 140 bp of the control (Fig 3A and B). These results show that RAP1 protects critically short telomeres in replicative senescence cells. However, this does not exclude that in other situations, e.g., in cancer cells, critically short telomeres are protected by RAP1.

**RAP1 protects critically short telomeres from fusions in HeLa cells**

Then, we asked whether the telomere protective role of RAP1 seen in senescent cells is indeed due to the presence of critically short telomeres and/or to the particular context of cellular senescence. Thus, we investigated the RAP1-protective role by using a HeLa cell line containing a doxycycline-inducible CRISPR knockout of RAP1 [26]. We treated cells for 25 days with the telomerase inhibitor BIBR1532, thus creating short telomeres, and with doxycycline for 15 days to achieve the depletion of RAP1 (Fig 4A). As expected, BIBR1532 treatment caused a decrease in telomere length (Figs 4B and EV3A) and similar to MRC-5 senescent cells, no further telomere shortening was observed when RAP1 was depleted (Fig EV3B). Upon RAP1 depletion, there is an increase in fusion frequency only in HeLa cells treated with BIBR1532 as measured by either the PCR-based strategy used in senescent cells (Fig 4C) or counting telomere abnormalities in telomere-stained metaphase spreads (Fig 4D and E). Notably, in metaphases, we did not observe an increase of telomere fragility (as recorded by multiple telomere signal (MTS)) nor telomere loss upon RAP1 downregulation.

Once again, and in agreement with the MRC-5 results, depletion of LIG4 but not of LIG3 prevents the appearance of fusions in RAP1-compromised cells (Figs 4C and EV3C and D). Finally, we asked whether the increase in fusions upon RAP1 depletion was due to a 3′ overhang trimming. However, we were unable to detect any change in the length of the overhang in HeLa cells upon RAP1 depletion and BIBR1532 treatment (Fig EV3E), indicating that RAP1

**Figure 3. Telomere fusions in senescent cells contain short telomere arrays.**

A   Distribution of the telomere repeat array found at the junction between two fused chromosomes of senescent cells transduced with lentiviral vectors expressing either shControl or shRAP1. Variant telomere repeats were included in the estimation of the telomere array length. Data represent median $\pm$ interquartile range (green lines) of three biological replicates. Statistical analysis was performed using Mann–Whitney $U$-test ***$P < 0.0001$).

B   Examples of DNA sequences found at the telomere fusion points of shControl and shRAP1 conditions. Telomere repeats are underlined in blue, while variant telomere repeats are underlined in red. The sequence ID corresponds to the identifier in Table EV1.

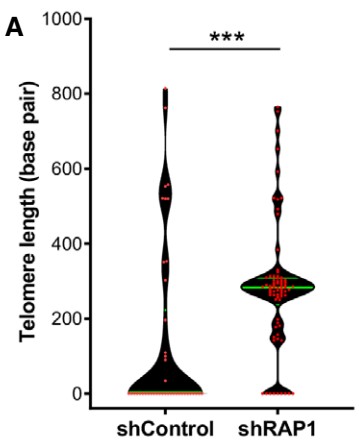

**A**

***

**B**

## shControl-PD72

Telomeric repeats
Variant telomeric repeats

Sequence ID: ebb72916-e66a-4693-8b2a-74db00e44ea6

Chr. 9 "+"

...GGCTGAGTGCAGTGGTGTGAGTCTCAGCTCACGCACCTCACCTCCCGGGTTCAAGTGATTCTC CTGCTCAGCCTACTGAGTAGTTGGGATTACAGGCGCCTGCCATGCCCGGCTAATGTTTGTATTT TTAGTAGAGACTGGGGTTTCACCATGTTGGCCAGCTGGTCTTGTGACAAGCCTCTGGTGATCTC GCACCTCGGTTTCCCTTGGGAGAGAGAGGGGGTTAGTGTTAGGGGTTTAGGGTTAGGGTTCGGG TTTGGGTTTGTAGTGTTAGGGCTAGGGGTTTTTGGGTTAGGAGAATTGCTTGAACCTGGGAGGTG GTGGTTGCAGTGAGCTGAGAGTGTGCCACTGCTCTCCAGCCCAGGCTGGAAACCAGACTCCATC TCAAAAAAAAAAAAAAAAAAAAAAAAAAAAAAGCAAACATGAGTGTTTAAACCTGAAATTTTTGCA ACTGGTATATGTGTGATATACACACACACATACACACACACACAGGAGCTGGAGAATATAGTAA GGTAATATTCAGC.....

Chr. 9 "-"

Sequence ID: ff6f2e6c-71d3-418a-aa3d-5b33510e767b

Chr. 6"-"

...GGTGGTCAGAGAGGTGAAAACTTTAGGCTGGGGATTTCCTTTATGAATTCTTAAATTTTCCATACACAAGGGTAACTTCTGCTGTTTTCAGAACTTCCT TTATTTAGCATTTATTTTTCAAAATAATGGCTTGGAATAATTCTTAAGACAAAGGGACATATTTTGGGGTGGCATATCACGGACTTTTCTACCATTATAT TTTGGGGTGGCATGTTTGATCTTGTACATTTGTGTTCCACCGGCAATGAAAAAAGAGTTTCTTGTTTCCTCAACCGTTTATTCATTTTTAGGAGTTTACA CGGTTCTAAAGATATAGACCAGCTGTGCTTTGCTATCTCATTGTGGTTTCAGTTCTCTCTGTGTCATTTGAGGCATCTTTTTGTATGTTTTACTTGCCAT CTATGGAAGATCTTCTTTGGTAGTGTCTGTTTCAGATGTAGGCTGCTCAGAGCCAAGCCACCGGCTGGAGGGAGGGGCTCCCAGCAGGTGCGGCTTTGTG GCCCTGGGAGAGCAGGTGGAAGATCAAACAGGCCATCGCTGCCGCAGGACCAGTGGATTGGCTAGGTGGGATCTCTGGGCTCAACAAGCCCTCTGGGTG GTAGGTGCAGAGAGGGGAGGGGGGCAGAGCCGCAGGCACAGCCAAGAAGGGCTGAAGAAATGGTAGGGCAGGGCAGCTGGTGATGTGGGCCCACCGGCCC CAGGCTCCCTGTCTCCCCCCAGGTGTGGCGGTACCAGGCATGCCCTTCCCCCCGGCATCAGGTCTCCCAGAGCTGCAGAAGACGACGGCCGACTTGGGTC ACACTCTTGTAGGTATGCAGTGTTGCAGGTGAGAGAGAGAGTCGACAGTGAGTGGGAGTGGCGTACCCCTATAGGAACTCTACCCCTAGACGTCTCCTGT CTCCTGGAGGAAGCTGATGCC...

Chr. 16"+"

## shRAP1-PD72

Sequence ID >33341423-8a90-4dae-ad8e-3ebf6242b09a

Chr. 2 "-"

..........AACACAAATGCAGCATTACAAACAGACATGACACCGAAAATATAACACACCCCATTGCTCATGTAACAAGCACCTGTAATGCTAATGCACTGCCTCA AAGCAAATATTAATATAAGATCGGCGAAATCCGCACACTGCCGTGCAGTGCTAAGACAACAATGAAAATAGTCAACATAATAACCCTAATAGTGTTAGGGT TAGGGTCAAGACCCCGGTCCGGGTCAGGGGTCAGGTCAGGTCCAGGTCAGGGTGAGGGTTAGGGTTAGGGCGAGGTTAGAGGTTAGGGCGAGGTTAGGGTT AGGGCGAGGGTTAGGGTGTGAGGGTGAGGTTAGGGTGAGGGTGAGGGTTAGGGTGAGGGTGAGGGGTTAGGGTGAGGTTAGAGGTTAGGGTGGGGTTACAG TTAGGGTGAGGGTGAGGTTAGGGTTAGGGTTAGGGTTAGGGTTAGGGTTAGGGTTAGGGTTTAGGGTGAGGGTTAGGGTTAGGGTTTAGGGTGAGGGTTAG GGTTAGGGTTAGGGTTAGGGTTAGGAGGGTTAGGGTTAGGGTTAGGGTTAGGGTTAGGGTTAGGGTTAGGGTTAGGGTTAGGGTTAGGGTTAGGGTTCG AGTTAGGGTTCGGGTTCGGGTTCGGGTTCCGGGTTCGGGTTCGGGTTCAGGGTTAGGAGGTTCGGGTTCGGAGTTAGGGTTAAAGTTAGGGTTAG GGTTAGGGTTAGGGTTAGGGTTTAGGGTTAGGGTTAAGGGTTTAGGGTTAGGGTTAGGGTTAGGGTTAGGGTTAGGGTTAGGGTTAGGGTTAGGGTTAGG GTTAGGGTTTAGGGTTAGGGTTCGGGTTAGGGTTCGGGTTAGGGTTCGAGGTTCGGGTTCGGGTTAGGGTTCGGGTTCGGGTTCAGGTTCGGGTTCGGGTT AAGTTGGGGTTCGGGTTAGGGTTAGGAGGTTAGGGTTGGGTCAGGGGCCCCGGGAGGCCGGACCTTTGGAGTG...

Chr. 12 "+"

Sequence ID >943281fb-42e7-4c24-9612-377afd142215

Chr. Y "-"

...AATTGGTGGGTTCTTGGTCTCACGCCGACTTCAAGGAATGAAGACATGGAACCTCGCGGTGAGTGTTACAGTTTCTTAGAATTGTGCGTCCGGGTTTGTT TCTTCTGATGTTCAGATGTGTGTTCTGAGTTTCTTCTTTCTGGTGGGGTTGGTCTCACTGGCTCAGGAGTGAGGCTGCAGACCTTTGCAGTGAGTGTCACA GCTCCTTAAAGGCAGTGTGGCCAAAGGTGAGCAATAGCAAGATTTATTGCAAGAGTGAAGAACAGAAGCTTCACAGTATGGAAAGGACTGTTGGGTTGCC ACTGCTAGCTCAGAGCAGTCTGCTTTTTATTCTCTAATCTGCTCCCACCCACATCCTGCTGATGGGTCCACTTTCAGAGGGTTAGGGTTAGGGTTAGGGTTA AGTTGGGGTTTAGGGTCAGGGTTAGGGGTTAGGGTCAGGGTCAGGGTCAGGGTCAGGGTCAGGGTCAGGGTCGGGATTCAGGGTCAGGGGTCAG GGGTCAGGGTCAGGGTTAGGGTCAGGGGTTAGGGTTTGGGTCAGGGTTAGGGTTAGGGTTAGGGTTGGGGTTAGGGTTAGGGTTAGGGTTAGGGTTAGGGT TAGGGTTAGGGTTAGGGTTAGGGTTGGGGTTAGGGGTTAGGGTTTTGGGTTAGGGTTAGAGGCGCGCCGCTGTTGGGGGAGACGCAGGCAGGGCGTAGACGCA CGCCGGCGCCTCCCGGAGAGGGGTCGCTGGGCAGGCAGGAGTGAGAGCGCAGCACCAGGCTCAGAGACGCACGTCGCTGGGCTGAGGTGGCGAAGTGTTGC AGTCACACAGTCATCGCCGCCGGGCGGGAACGCGGGGGTGGCGCAGTGCACGCGCGCGCAGAGACACGTCCCCGGCGGCGCAGAGACGAGTGGGACCTGA GTAATCTGAAAAGCCCGTTTCGGGCGCCCCTGCTTGCAGCCGGGCACTACGGGACCAGCTTGCCCACGGTGCTCTGCCATTGCGCTACTGGCGACTAGGAC AGCTGCAGGGCCCTCTTGCTTACAGTGGTGTCCAGCGCCCCGTGCTGGCGCCCAGGGCACGGCAGGAGCTCTCTTGCTCGCAGTATAGTGGTGGCACGCCT GCTGGCAACTAGGGACATTGGGGCCCTCTTCCTCACATTATAGTGGCAGCACACCCGCCTGCGCGGCGCTGGGCACACTGCCGGACCCTCTTG.....

Chr. 10 "+"

**Figure 3.**

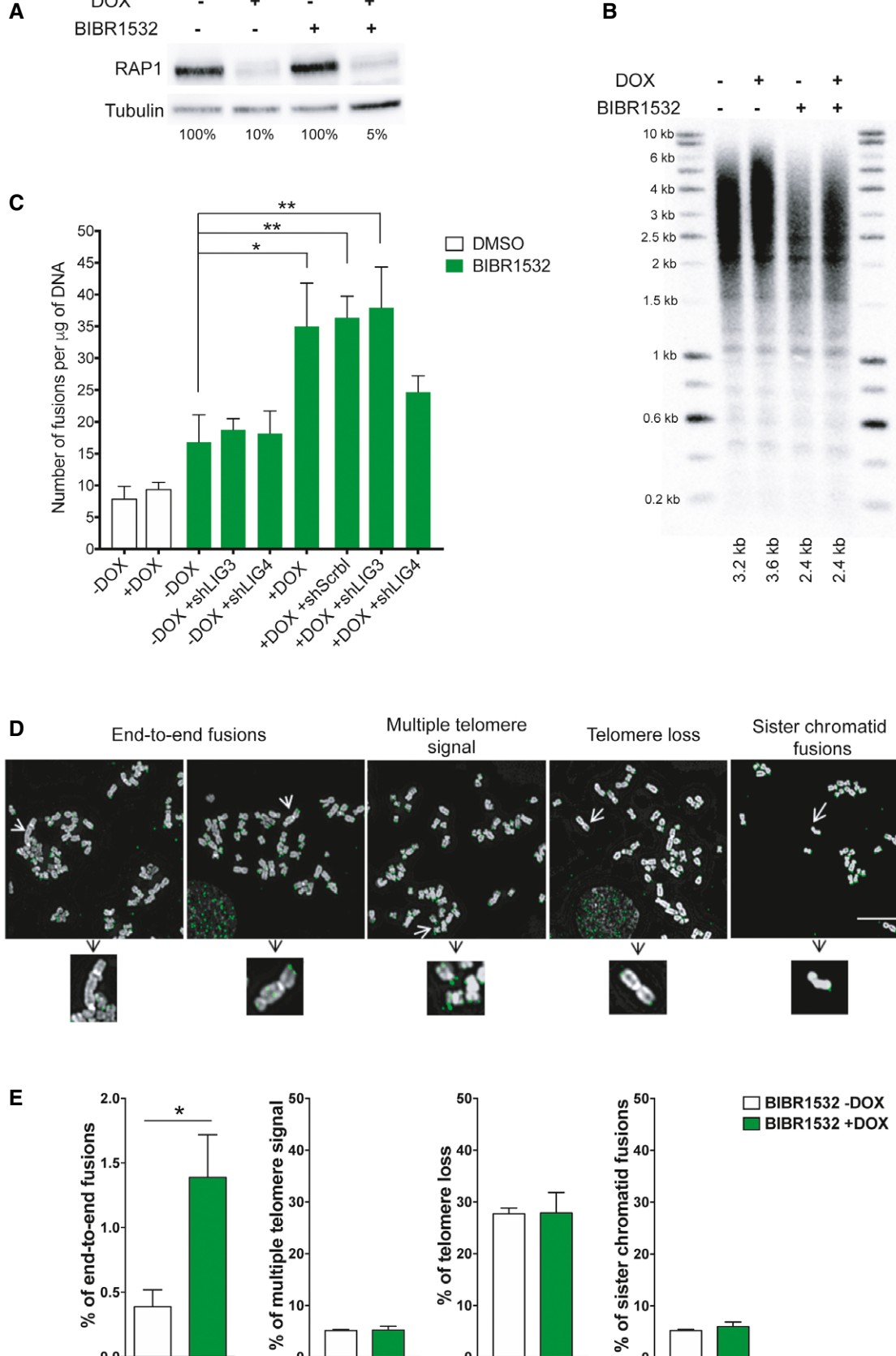

Figure 4.

**Figure 4. Telomere fusions in HeLa cells after RAP1 depletion.**

A  RAP1 expression in HeLa cells treated with the telomerase inhibitor BIBR1532 for 25 days (20 μM final concentration). Fifteen days before cell harvesting, doxycycline (DOX; 1 μg/μl final concentration) was added to deplete the expression of RAP1.

B  Telomere length analysis by Southern blotting of the samples described in (A). The size of the main intensity peak is indicated at the bottom of the gel.

C  Number of fusions in HeLa cells after 25 days in culture. Cells were maintained with BIBR1532 during the whole period of the experiment, while doxycycline (1 μg/μl final concentration) and the infection with shControl, shLIG3, or shLIG4 were carried out for the last 15 days of the experiment. Data represent mean ± SD of three biological replicates. Statistical analyses were performed using Mann–Whitney $U$-test (*$P < 0.05$; **$P < 0.001$; ***$P < 0.0001$).

D  Examples of metaphase spreads hybridized with a telomeric PNA probe (green) when RAP1 was depleted (+DOX). Telomerase was inhibited with BIBR1532 for 25 days. Scale bar = 10 μm.

E  Quantification of chromosome aberrations observed in RAP1-depleted cells (+DOX) or control (-DOX). Data represent mean ± SD of three biological replicates. Approximately 15 metaphase spreads were analyzed per replicate with a total of 2,300 chromosomes examined per condition. *$P < 0.05$; two-tailed Student's $t$-test.

does not change the regulation of the 5′ end resection to prevent fusions. However, we cannot rule out telomere overhang loss after RAP1 depletion of only a subset of chromosomes containing very short telomeres.

Martinez and colleagues [11] showed that RAP1 has a protective role on mouse telomeres in Terc$^{-/-}$ mice. In contrast to our cellular settings, telomere shortening was accelerated in the double knockout mice (Terc$^{-/-}$, Rap$^{-/-}$) as compared to Terc$^{-/-}$, giving rise to an increased number of telomere aberrations such as signal-free ends, MTS, telomere sister chromatid exchanges, and end-to-end fusions. Thus, together with our results, mammalian RAP1 appears to have the capability to protect against telomere fusions *in vivo* by different mechanisms including NHEJ inhibition between critically short telomeres (this study) and prevention of fusogenic telomere-free ends [7].

Overall, our results indicate that human RAP1 is required to protect critically short telomeres from classical NHEJ-mediated fusions in senescent or cancer cells. Previously, we showed that HeLa cells expressing a TRF2 mutant named "Top-less" that is unable to form t-loops activate the ATM checkpoint and rely on RAP1 to protect telomeres from fusions [12]. Altogether, we propose that RAP1 is required to protect telomeres that are too short to be able to fold into a t-loop but still capable to bind TRF2. This model is further supported by the fact that both TRF2 and RAP1 are detectable at telomeres of senescent cells (Fig 1B). Interestingly, RAP1 in senescent cells does not only protect against telomere fusion but also against 53BP1 recruitment to telomeres (Fig 1C). Since 53BP1 promotes NHEJ by increasing chromatin mobility [27,28] and/or by preventing end resection thanks to its interaction with RIF1 and the shieldin complex [29,30], it is tempting to propose that the anti-fusion properties of RAP1 rely on its ability to prevent 53BP1 binding to telomeres.

## Reversal of senescence is compromised upon RAP1 depletion

A prediction of the above results is that cells bypassing the replicative senescence checkpoint would strongly rely on RAP1 for their telomere protection. In order to address this point, we took advantage of the fact that the cell-cycle arrest of replicative senescent MRC-5 cells can be reversed by inhibiting the p53-p21$^{CIP1}$ tumor suppressor pathway [31]. Thus, we infected senescent MRC-5 cells with either an shRNA against p21$^{CIP1}$ (shp21$^{CIP1}$) or shp21$^{CIP1}$ + shRAP1 (Fig 5A and B). Two weeks post-infection, about 66% of the shp21$^{CIP1}$-transduced cells were dividing as compared to only 9% in shp21$^{CIP1}$ + shRAP1 cells (Fig 5C). Even more, around 92% of the cells were positive for senescence-associated β galactosidase (SA-β-gal) staining (Fig 5D). However, as can be appreciated in the growth curve (Fig 5A), shp21$^{CIP1}$ + shRAP1 cells undergo high levels of cell death. In order to investigate this further, we measured the levels of apoptosis in our cultures (Fig 5E and F). Interestingly, RAP1 depletion in cells that return to growth caused significantly higher levels of apoptosis compared to control cells (Fig 5E and F), more likely by the excess of telomere and chromosome instability.

Overall, our results together with the fact that RAP1 is a highly conserved telomere protein raise the question of the physiological role of the protection properties of RAP1 toward critically short telomeres. One possibility is that RAP1 guarantees the stability of critically short telomeres that may occur on stochastically basis during cell division [32]. In senescent cells or cells approaching senescence, it might also be important to prevent telomere fusion in order to reduce the risk of aberrant chromosome rearrangement in the event of checkpoint bypass.

**Figure 5. RAP1 depletion in post-senescent MRC-5 cells.**

A  Growth curve of MRC-5 fibroblasts. Cells were grown until senescence and after 3 weeks of no growth detection, lentiviral infections containing either shp21$^{CIP1}$ or shp21$^{CIP1}$ + shRAP1 sequences were performed. Cells were harvested 15 days post-infection (dpi). Data represent mean ± SD of three biological replicates measured after 8 and 15 dpi.

B  Expression of p21$^{CIP1}$ and RAP1 after 15 dpi of cells transduced with shControl, shp21$^{CIP1}$, or shp21$^{CIP1}$ + shRAP1.

C  Dividing cells (magenta) were identified by EdU staining (1 μM EdU for 24 h). The percentage of dividing cells after 15 dpi is shown ($n = 3$). Scale bar = 10 μm.

D  Senescence-associated β-galactosidase (SA-β-gal) staining in senescent and post-senescent fibroblasts (1 and 15 dpi) is shown. Percentage of SA-β-gal-positive cells is indicated for each condition. Three biological replicates were performed, with approximately 300 cells analyzed per condition. Scale bar = 50 μm.

E  Senescent cells, transduced with shp21$^{CIP1}$ or shp21$^{CIP1}$ + shRAP1 for 15 days, were stained with CytoCalcein violet (live cells), Apopxin Green (apoptotic cells), and 7-aminoactinomycin D (7-AAD) (late apoptotic/necrotic cells) and visualized by flow cytometry.

F  Quantification of apoptotic cells from the conditions described in (E). Data represent mean ± SD of three biological replicates (**$P < 0.001$; two-tailed Student's $t$-test).

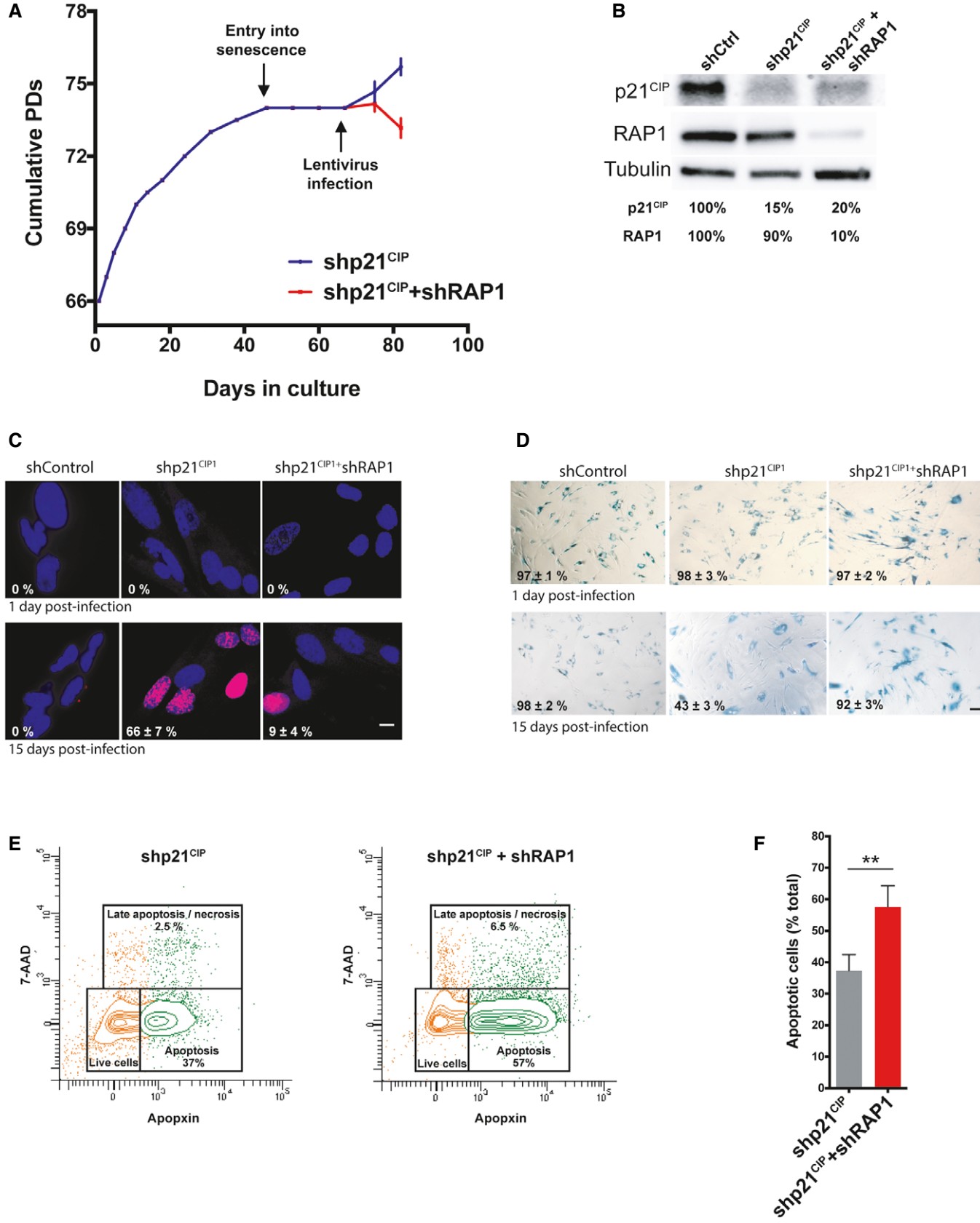

**Figure 5.**

# Materials and Methods

### Cell lines and reagents

MRC-5 human primary lung fibroblasts were obtained from ATCC. MRC-5 cells were grown in DMEM supplemented with 10% fetal bovine serum, penicillin (100 IU/ml), and streptomycin (100 μg/ml). Cells were cultured at 37°C, 5% $CO_2$, 5% $O_2$. HeLa CRISPR/Cas9 engineered cell line with the doxycycline-inducible knockout of *RAP1* was a gift from Dr. Songyang [26]. Cells were grown in DMEM with 10% tetracycline-free serum. To induce knockout of RAP1, cells were treated for 15 days with doxycycline (2 μg/ml, Sigma). Telomere shortening was selectively induced by treatment with the telomerase inhibitor BIBR1532 (20 μM, Merck).

### Lentivirus infection and siRNA transfection

Lentiviruses were produced by transient calcium phosphate transfection of 293T cells with the virus packaging plasmids, p8.91 and pVSVg, as well as with the lentiviral expression vector that contained the sequence of interest. Titration was performed approximately 10 days after infection by means of puromycin (1 μg/ml) selection of clones.

The following shRNA plasmids were purchased from Sigma and used for lentivirus production: pLKO-shControl, pLKO-shTERF2IP, pLKO-shp21$^{CIP1}$, pLKO-shLIG3, and pLKO-shLIG4 (sequences below).

Several truncated RAP1 proteins were designed using the full-length pWPIR-RAP1WT-GFP construct as a template. pWPIR-RAP1ΔRCT-GFP (Δ271–330 amino acids) and pWPIR-RAP1ΔMYB-GFP (Δ132–183 amino acids) synthesis was performed by GenScript (Piscataway, USA).

Infection with various shRNAs or overexpression vectors was performed for a minimum of 4 days, and depending on the experiment, cells were kept in culture for up to 10–15 days after infection (10 days for telomere fusion assay in MRC-5, 8–15 days—for post-senescent cells).

siRNAs were purchased from Dharmacon, and transfections were performed with DharmaFECT1 transfection reagent (Dharmacon) for 6 days by performing 2 sequential transfections of 3 days each. Efficiency of each shRNA and siRNA was checked routinely by RT–qPCR or Western blotting.

Sequences of shRNA and siRNAs used in this work were as follows:
shLIG3: CCGGCCGGATCATGTTCTCAGAAATCTCGAGATTTCTGAGAACATGATCCGGTTTTTG
shLIG4: CCGGTTCGACGCCACACCGTTTATTCTCGAGAATAAACGGTGTGGCGTCGAATTTTTG
shRAP1: CCGGAGAGTATGTGAAGGAAGAAATCTCGAGATTTCTTCCTTCACATACTCTTTTTTG
shp21: CCGGCCGCGACTGTGATGCGCTAATCTCGAGATTAGCGCATCACAGTCGCGGTTTTTG
siRAP1: GAUGAGAGCCCUCCUGAUU
siLIG4: GCACAAAGAUGGAGAUGUA
siLIG3: GGACUUGGCUGACAUGAUA

### Antibodies

The following antibodies were used for Western blotting: rabbit polyclonal anti-RAP1, 1:5,000 (Bethyl Laboratories, Inc., A300-306A), rabbit polyclonal anti-TRF2, 1:5,000 (Novus Biologicals, NB110-57130), mouse monoclonal anti-α-tubulin, 1:2,000 (Merck, T9026), rabbit polyclonal anti-GAPDH, 1:1,000 (Novus Biologicals 100-56875), mouse monoclonal anti-p21, 1 μg/ml (Abcam, ab16767), rabbit polyclonal anti-DNA ligase III 1:500 (Abcam, ab185815), rabbit polyclonal anti-DNA ligase IV 1:1,000 (CST, 14649S), HRP goat anti-mouse IgG, 1:10,000 (Vector Laboratories, PI-2000), and HRP goat anti-rabbit IgG, 1:10,000 (Vector Laboratories, PI-1000). Antibodies used for immunofluorescence were as follows: rabbit polyclonal anti-53BP1, 1:250 (Novus Biologicals, NB100-305) and goat anti-rabbit Alexa 488 antibody, 1:400 (111-545-144; Jackson ImmunoResearch). Antibodies used for ChIP were as follows: rabbit polyclonal anti-TRF2 (Novus Biologicals, NB110-57130) and rabbit polyclonal anti-RAP1 (Bethyl Laboratories, Inc., A300-306A), IgG control (Merc, I-5006).

### Primers

The following primers were used for RT–qPCR: TERF2IP-F: CGGGGAACCACAGAATAAGA, TERF2IP-R: CTCAGGTGTGGGTGGATCAT, 36B4-F: AACTCTGCATTCTCGCTTCCT, 36B4-R: ACTCGTTTGTACCCGTTGATG, p21$^{CIP1}$-F: TGGTAGGAGACAGGAGACCT, p21$^{CIP1}$-R:AATACTCCCCACATAGCCCG, LIG3-F: GAT CAC GTG CCA CCT ACC TTG T, LIG3-R: GGC ATA GTC CAC ACA GAA CCG T, LIG4-F: CAC CTT GCG TTT TCC ACG AA, and LIG4-R: CAG ATG CCT TCC CCC TAA GTT G.

Primers used for telomere fusion assay were as follows (described in [24]): 21q1: 5′-CTTGGTGTCGAGAGAGGTAG-3′, 16p1: 5′-TGGACTTCTCACTTCTAGGGCAG-3′, and XpYpM: 5′-ACCAGGTTTTCCAGTGTGTT-3′.

Primers for generation of subtelomeric DNA probes were as follows: XpYpO: 5′-CCTGTAACGCTGT TAGGTAC-3′, XpYpG: 5′-AATTCCAGACACACTAGGACCCTGA-3′, 21qseq1: 5′-TGGTCTTATACACTGTGTTC-3′, 21qseq1rev: 5′-AGCTAGCTATCTACTCTAACAGAGC-3′, 16p2: 5′-TCACTGCTGTATCTCCCAGTG-3′, and 16pseq1rev: 5′-GCTGGGTGAGCTTAGAGAGGAAAGC-3′.

Primers used for STELA were as follows (described in [33]): XpYpE2: TTGTCTCAGGGTCCTAGTG, telorette: TGCTCCGTGCATCTGGCATCTAACCCT, and teltail: TGCTCCGTGCATCTGGCATC.

### RNA extraction and RT–qPCR

Total RNA was extracted following instructions of the RNeasy Mini Kit (Qiagen), and then, 1 μg of RNA was reverse-transcribed into cDNA using the High-Capacity RNA-to-cDNA Kit (Thermo Scientific). Each qPCR contained 10× diluted cDNA, 0.2 μM primers, and SYBR Green Master Mix (Roche, 4913914 001).

### DNA extraction

Genomic DNA was extracted either by proteinase K, RNase A, and phenol/chloroform followed by ethanol precipitation or following instructions of DNA Blood and Tissue Kit (Qiagen).

### Chromatin immunoprecipitation (ChIP)

Cells were cross-linked for 12 min with 1% formaldehyde and then washed with cold PBS. The cells were centrifuged, and the pellet re-suspended in cell lysis buffer (5 mM PIPES pH 8, 85 mM

KCl, 0.5% NP40, and protease inhibitors). Cells were pelleted again and re-suspended in nucleus lysis buffer (50 mM Tris pH 8, 10 mM EDTA, 1% SDS, protease inhibitors). Cells were sonicated using a Bioruptor, and IPs were set up with around 30 μg of DNA per condition with an overnight incubation of the desired antibody. Magnetic beads (Dynabeads, Life Technologies) were added for 2 h. The beads were washed with a low salt buffer (150 mM NaCl, 1% Triton X-100, 0.1% SDS), a high salt buffer (500 mM NaCl, 1% Triton X-100, 0.1% SDS), followed by a lithium salt buffer (0.25 M LiCl, 1% NP40, 1% deoxycholic acid). Chromatin was eluted (1% SDS, 0.1 M NaHCO$_3$ solution), and the cross-linked chromatin was reversed at 65°C overnight. The DNA was treated with RNaseA (10 mg/ml for 20 min) and proteinase K (10 mg/ml for 1 h at 50°C), followed by phenol–chloroform purification and ethanol precipitation. The DNA obtained from ChIP was denatured and blotted onto nylon membranes using a slot blot apparatus, cross-linked, and hybridized with different radioactively labeled probes. The membranes were exposed onto phosphorimager screens, and the signal intensity was quantified with ImageQuant software.

## Telomere fusion assay

We performed telomere fusion assay as described before [23,24] with some modifications. Shortly, genomic DNA was digested with EcoRI (Promega), and accurate concentrations were measured using Qubit fluorometer (Thermo Fisher Scientific). PCRs were performed with a mix of three subtelomeric primers (21q1, 16p1, and XpYpM; 0.2 μM final concentration each) and the FailSafe™ PCR System (Lucigen) under the following conditions: 26 cycles of 94°C for 15 s, 58°C for 30 s, and 68°C for 10 min. PCR products were resolved on 0.8% agarose gel followed by Southern blotting. Each nylon membrane was subsequently hybridized with the XpYp, 16p and 21q radioactively labeled (αP32 dCTP) probes and a DNA ladder probe (SmartLadder MW-1700-10, Eurogentec). The membranes were exposed and revealed on the Typhoon FLA 9500 Phosphorimager (GE Healthcare Life Sciences). Molecular size of each of the bands was calculated by ImageQuant TL 8.1 software (GE Healthcare Life Sciences).

To estimate the frequency of fusions, the number of PCR bands of each DNA analyzed was counted and the total amount of DNA used in the PCRs was calculated. The frequency of fusions was expressed as number of fusion events per 1 μg of DNA used in the PCR. Only bands of different sizes after sequential hybridization with the three different subtelomeric probes were considered to estimate the frequency of fusions. Each PCR was carried out with 50 ng of EcoRI digested DNA, and each biological replicate was performed with 16 PCRs. The frequency of fusions represents an underestimate of the total telomere fusion in a given cells as: (i) The primers used in the PCR target only a subset of chromosome ends, (ii) there is no correction for PCR efficiency, and (iii) only fusion products of about 10 kb or lower are detected with the conditions used in this study.

## STELA

We performed STELA as it was described in Ref [33] with some modifications. Briefly, total genomic DNA was digested with EcoRI, quantified by Qubit fluorometer (Thermo Fisher Scientific). A ligase

reaction was performed with 10 ng DNA and the telorette linker oligo with 0.5 units of T4 DNA ligase (Promega). Ligated DNA (250 pg) was used for PCR with telomere-adjacent (XpYpE2) and teltail primers, and the FailSafe™ PCR System (Lucigen). We cycled the reactions under the following conditions: 26 cycles of 94°C for 15 s, 62°C for 30 s, and 68°C for 10 min. PCR products were resolved on 0.8% agarose gels followed by Southern blotting. Hybridized membranes were exposed and the signal detected on the Typhoon FLA 9500 (GE Healthcare Life Sciences). Analysis was performed using the ImageQuant TL 8.1 software (GE Healthcare Life Sciences).

## Nanopore sequencing

The PCR products, generated by the telomere fusion assay (shControl and shRAP1 of MRC-5 senescent cells), were cleaned up with standard phenol–chloroform followed by ethanol precipitation procedures. One μg of purified DNA (per condition) was used for the preparation of libraries. DNA repair and end preparation were performed using the NEBNext FFPE DNA Repair Mix with the following reaction setup: 48 μl DNA, 3.5 μl NEBNext FFPE DNA Repair Buffer, 2 μl NEBNext FFPE DNA Repair Mix, 3.5 μl Ultra II End Prep Reaction Buffer, and 3 μl Ultra II End Prep Enzyme Mix; 20°C for 15 min followed by 65°C for 15 min. Afterward, the DNA size selection was carried out using AMPure XP Beads (NEB) followed by native barcode ligation (22.5 μl DNA, 2.5 μl native barcode provided by EXP-NBD104 kit and 25 μl Blunt/TA Ligase Master Mix; 25°C for 20 min). After another round of AMPure XP bead clean-up, the samples were pooled together and adaptors were ligated for the pooled sample at 25°C for 15 min (65 μl DNA, 5 μl AMII provided by EXP-NBD104 kit, 20 μl NEBNext Quick Ligation Reaction Buffer, and 10 μl Quick T4 DNA Ligase). The adaptor-ligated DNA was cleaned up by adding a 0.4× volume of AMPure XP beads followed by incubation for 5 min at room temperature and re-suspension of the pellet twice in 140 μl ABB solution (SQK-LSK108 kit). The purified ligated DNA was re-suspended in 15 μl Elution Buffer (SQK-LSK108 kit) and incubated with the beads for library preparation at room temperature for 10 min. The solution was centrifuged, and the supernatant was transferred to a new tube for library loading. The library was loaded on the FLO-MIN106 MinION flow cell according to the manufacturer's guidelines (SQK-LSK108 kit). MinION sequencing was performed by Oxford Nanopore Technologies MinKNOW software.

## Nanopore sequencing analysis

The raw nanopore reads were processed using the 00.Long_Reads module of the LRSDAY v1.5.0 framework [34]. Briefly, we performed base calling and debarcoding with Guppy v2.3.5. The debarcoded reads were further processed by Porechop v0.2.4 (https://github.com/rrwick/Porechop) for adapter trimming (option: –discard_middle). For each sample, the trimmed reads were mapped (option: -ax map-ont) to the human reference genome (version GRCh38) with minimap2 v2.16 [35]. The resulting BAM file was further processed by SAMtools v1.9 [36] and picard tools v2.18.23 (https://broadinstitute.github.io/picard/) for alignment sorting, filtering (minimal mapping quality cutoff: 30), and duplicates removing. Chimeric reads (i.e., different parts of the same read were uniquely mapped to different genomic regions) were extracted by SAMtools by only keeping the reads with the "SA:" SAM format

flag. The resulting chimeric reads were further filtered to only keep those mapped to the 20-kb most terminal regions of the human chromosomes (after removing the polyNs at the ends of the corresponding chromosomes). A final round of manual inspection was performed to verify the chimeric nature of these reads by searching them against the human reference genome using BLAT v36 (Kent 2002 Genome Research) and verified the matched regions (see Table EV1 for more details).

### Telomere length analysis by Southern blotting

Total DNA was digested with HinfI/RsaI enzymes (Promega), and 5 µg was migrated on 1% agarose gels. After transfer of DNA to the $N^+$ Hybond membrane (GE Healthcare), each membrane was hybridized with the telomeric DNA probe (purified 650-bp telomeric fragment) obtained by random priming using the Klenow large fragment enzyme and radioactively labeled (αP32) dCTP nucleotides. The signal was later revealed using the Typhoon FLA 9500 (GE Healthcare Life Sciences). The signal profile and the telomere length of the highest intensity peak were obtained using the ImageQuant RL 8.1 software (GE Healthcare Life Sciences).

### Telomere overhang assay

The overhang assay was performed as described by Ref [12]. Briefly, 5 µg of genomic DNA from HeLa cells was digested with HinfI and RsaI (Promega), and incubated with 0.2 pmoles of the radioactively end-labeled single-stranded $(CCCTAA)_3$ probe overnight at 50°C. Hybridized samples were loaded into 0.9% agarose gels and size separated by electrophoresis (6 V/cm for 75 min). The gel was then dried on 3-mm paper for 4 h at 40°C and exposed on a phosphorimager screen. Telomere overhang signal was normalized to the total telomere signal, which was obtained by hybridization of the same gel but after denaturation and using the same probe as the native gel. Analysis was performed using the Typhoon FLA 9500 (GE Healthcare Life Sciences).

### Western blotting

Protein extracts were obtained by lysis in ice-cold RIPA buffer for 30 min followed by 30-min centrifugation at 4°C. Proteins were separated on 4–20% acrylamide gradient SDS gels (Bio-Rad) and transferred to Amersham Protran 0.45-µm nitrocellulose membranes (GE Healthcare Life Sciences) for 90 min at 300 mA. Further, the membranes were blocked in 5% skim milk in PBST buffer and incubated thereafter with the primary and secondary antibodies. Membranes were developed using the Luminata Forte HRP substrate (Millipore) and exposed in the Fusion Solo apparatus (Vilber Lourmat).

### IF-FISH

Immunofluorescence–FISH in MRC-5 cells and further analysis of the images were performed as described in Ref [37]. Shortly, cells were grown on coverslips, and after determined by the specific assay time, they were fixed in 4% formaldehyde followed by permeabilization in 0.5% Triton X-100. Afterward, FISH was performed: Cells were dehydrated in ethanol and further incubated for at least 2 h with the denatured telomeric PNA probe Cy3-OO-CCCTAACCCT

AACCCTAA. Post-hybridization washes were performed with 70% formamide, 10 mM Tris pH 7.2, followed by wash with 50 mM Tris pH 7.5, 150 mM NaCl, 0.05% Tween-20. Next, cells were blocked in 3% BSA, 0.3% Triton X-100, and hybridized subsequently with primary and secondary antibodies mentioned above.

### Metaphase chromosome analysis

To obtain chromosome spreads, HeLa cells, treated selectively with BIBR1532 and doxycycline as described above, were arrested in metaphase using 50 ng/ml colcemid (KaryoMAX, Invitrogen) for 2 h at 37°C. Afterward, trypsinized cells were incubated with hypotonic solution (75 mM KCl) for 15 min at 37°C, fixed with ice-cold methanol: glacial acetic acid (3:1), and spread on slides. FISH with the telomeric PNA probe was performed as described above. Stained metaphase chromosomes were visualized on the Zeiss Axiovert Z2 epi-fluorescent microscope and analyzed using the metasystem ISIS software.

### Apoptosis and senescence-associated β-galactosidase (SA-β-gal) assay

The detection of apoptotic cells was performed using the Apoptosis/ Necrosis Assay Kit (Abcam, ab176749), while the detection of SA-β-gal-positive cells was carried out using the Senescence Detection Kit (Abcam, ab65351) following the manufacturer's instructions.

### EdU staining

Click-iT® EdU Imaging Kit (Thermo Scientific) was used to score the percentage of proliferating MRC-5 cells. Images were taken with the DeltaVision widefield deconvolution microscope (GE Healthcare).

### Statistics

Statistical analysis was performed using the Prism 5 software (GraphPad). For comparison of two groups, we used two-tailed Mann–Whitney $U$-test or Student's $t$-test, and for multiple groups, the Kruskal–Wallis test was used. $P < 0.05$ were considered significant (*$P < 0.01$, **$P < 0.001$, ***$P < 0.0001$).

## Data availability

Nanopore sequencing data generated in this study are available in the Sequence Read Archive (SRA) repository (https://www.ncbi. nlm.nih.gov/sra/PRJNA577013), accession code: PRJNA577013.

Expanded View for this article is available online.

### Acknowledgements

This work was performed using the genomic, PICMI, and CYTOMED facilities of IRCAN (supported by FEDER, ARC, le Minitère de l'Enseignement Supérieur, la région Provence Alpes-Côte d'Azur, and Inserm). The exchanges between the EG and JY laboratory were supported by the PHC Cai Yuanpei program of the Ministry of Foreign Affair. The work in EG laboratory was supported by grants from the Fondation ARC ("Programme Labellisé"), ANR (program TELOCHROM) Institut Nationale du Cancer (INCa) (program REPLITOP), "Investments for the Future" LABEX SIGNALIFE (reference ANR-11-LABX-0028-01), FRM grant

(FDT20170437327), and the cross-cutting program of Inserm on aging (AGEMED). The work in the JY laboratory was supported by the National Natural Science Foundation of China (grant numbers 81522017, 81971312, and 91749126), the Shanghai Municipal Education Commission (Oriental Scholars Program), the Shanghai Municipal Science and Technology Commission (19XD1422500), and the Foundation of Shanghai Jiaotong University School of Medicine for Translational Medicine Innovation Project (grant number 15ZH4005).

## Author contributions

LL and AM-B conducted most of the experiments; J-XY, JL, and GL performed and analyzed Nanopore sequencing; MJG-P conducted telomere overhang assay and revised manuscript; ZS provided the HeLa CRISPR cellular system; NJR helped to set up the PCR-fusion assay; JY, EG, and AM-B coordinated the study and designed experiments; and LL, EG, and AM-B wrote the manuscript.

## Conflict of interest

The authors declare that they have no conflict of interest.

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
