## [Review Process File · EMBO Reports]

Human RAP1 specifically protects telomeres of senescent cells from DNA damage

Lototska L, Yue JX, Li J, Giraud-Panis MJ, Songyang Z, Royle NJ, Liti G, Ye J, Gilson E, Mendez-Bermudez A

Review timeline:

Submission date:	14 August 2019
Editorial Decision:	17 September 2019
Revision received:	26 November 2019
Editorial Decision:	23 January 2020
Revision received:	26 January 2020
Accepted:	29 January 2020

Editor: Esther Schnapp

Transaction Report:

1st Editorial Decision

17 September 2019

Thank you for the submission of your manuscript to EMBO reports. We have now received the enclosed referee reports on it.

As you will see, all referees acknowledge that the data are potentially interesting. However, referees 2 and 3 also point out a number of technical issues that preclude a solid interpretation of the experimental evidence provided, including the lack of statistics and controls. I think all referee concerns are reasonable should therefore be addressed. Please let me know if you disagree and we can discuss the revisions further.

I would thus like to invite you to revise your manuscript with the understanding that the referee concerns must be fully addressed and their suggestions taken on board. Please address all referee concerns in a complete point-by-point response. Acceptance of the manuscript will depend on a positive outcome of a second round of review. It is EMBO reports policy to allow a single round of major revision only and acceptance or rejection of the manuscript will therefore depend on the completeness of your responses included in the next, final version of the manuscript.

Revised manuscripts should be submitted within three months of a request for revision; they will otherwise be treated as new submissions. Please contact us if a 3-months time frame is not sufficient for the revisions so that we can discuss this further. You can either publish the study as a short report or as a full article. For short reports, the revised manuscript should not exceed 27,000 characters (including spaces but excluding materials & methods and references) and 5 main plus 5 expanded view figures. The results and discussion sections must further be combined, which will help to shorten the manuscript text by eliminating some redundancy that is inevitable when discussing the same experiments twice. For a normal article there are no length limitations, but it should have more than 5 main figures and the results and discussion sections must be separate. In both cases, the entire materials and methods must be included in the main manuscript file.

Regarding data quantification, please specify the number "n" for how many independent experiments were performed, the bars and error bars (e.g. SEM, SD) and the test used to calculate p-values in the respective figure legends. This information must be provided in the figure legends. Please also include scale bars in all microscopy images.

3) We replaced Supplementary Information with Expanded View (EV) Figures and Tables that are collapsible/expandable online. A maximum of 5 EV Figures can be typeset. EV Figures should be cited as 'Figure EV1, Figure EV2' etc... in the text and their respective legends should be included in the main text after the legends of regular figures.

5) a complete author checklist, which you can download from our author guidelines <https://www.embopress.org/page/journal/14693178/authorguide>. Please insert information in the checklist that is also reflected in the manuscript. The completed author checklist will also be part of the RPF.

6) Please note that all corresponding authors are required to supply an ORCID ID for their name upon submission of a revised manuscript (<https://orcid.org/>). Please find instructions on how to link your ORCID ID to your account in our manuscript tracking system in our Author guidelines <https://www.embopress.org/page/journal/14693178/authorguide#authorshipguidelines>

7) We would also encourage you to include the source data for figure panels that show essential data. Numerical data should be provided as individual .xls or .csv files (including a tab describing the data). For blots or microscopy, uncropped images should be submitted (using a zip archive if multiple images need to be supplied for one panel). Additional information on source data and instruction on how to label the files are available at <https://www.embopress.org/page/journal/14693178/authorguide#sourcedata>.

I look forward to seeing a revised version of your manuscript when it is ready. Please let me know if you have questions or comments regarding the revision.

REFEREE REPORTS

Referee #1:

This is a very interesting paper shedding new light on the function of the TRF2 co-factor RAP1 and on telomere protection in human senescent cells, two key issues of telomere and cancer biology.

The authors convincingly show that RAP1 loss leads to a significant increase in 53BP1- LIG4-dependent telomere fusions specifically in senescent cells, not in dividing primary cells. This response is remarkably restricted to NHEJ since RAP1 loss does not result in other telomere defects such as telomere fragility or telomere loss. Finally, RAP1 loss compromises cell viability when senescent cells are allowed to return to growth, a possible consequence of the increased frequency of chromosome fusions.

These results have important implications in the field of genome stability; hence, the paper will be read with interest by a wide audience.

Minor points:

1. The claim that RAP1 loss has no consequences in non-senescent dividing cells may be an overstatement. All the assays used here have a limited sensitivity and a significant background level. Caution regarding this point is necessary.
2. Page 7, 2nd paragraph: 53BP1 also promotes NHEJ through the Rif1 pathway.

Referee #2:

In this manuscript entitled "Human Rap1 specifically protects telomeres of senescent cells from DNA damage," Lototska et al. aim to establish the telomere-binding protein Rap1 as an essential factor for the protection of short telomeres in cells undergoing replicative senescence. The authors argue that Rap1 maintains telomere integrity of senescing cells by inhibiting recruitment of the DNA damage factor 53BP1 to telomeres and preventing telomere-telomere fusions via classical nonhomologous end joining in a LIG4-dependent manner. The authors argue that Rap1 protects short telomeres in senescing MRC-5 primary cells lacking telomerase expression and in HeLa cells in which telomerase is inhibited chemically by BIBR1532. Lastly, the authors claim that Rap1 is necessary for senescent MRC-5 cells to re-enter the cell cycle and begin proliferation upon inhibition of the checkpoint protein p21.

Unfortunately, in its presented form, the data does not support the claim that Rap1 has a protective role at telomeres in aging cells. Many of the data presented is based on a single experiment and lacks appropriate statistical measures in order to interpret the results and make confident conclusions. In addition, several experiments lack fundamental (and critical) controls. The whole manuscript revolves on the use of a single shRNA. Minimally, the authors should use multiple RAP1 shRNAs to control for off-target effects. The key conclusions need to be strengthened by the use of clonal cell lines that have been genetically deleted of Rap1 (Rap1 Δ/Δ) and have been described in the literature. Overall, due to major concerns with the experimental data such as statistical rigor, lack of vital controls, and overinterpretation of minor phenotypic differences, this study is not suitable for publication. Specific concerns are as follows:

Concerns:

Figure 1

- 1) Figure 1A: Previously, it has been established that TRF2 is necessary to maintain Rap1 protein and that loss of TRF2 leads to a loss of Rap1 (Celli et al., NCB). The authors report that TRF2 levels drop precipitously at late PD's; however, Rap1 levels remain mostly unchanged. A comment for this discrepancy is requested from the authors.
- 2) Figure 1C: only one shRNA is used against Rap1. A second independent shRNA is necessary to control for potential off-target effects. This same critique applies to Figures 2 and 4.
- 3) Figure 1C: a rescue experiment should be performed by, expressing a Rap1 cDNA that is resistant to shRNA and showing a subsequent decrease of 53BP1 telomere foci to shScrb1 levels. This same critique applies to Figure 2C in the shLIG3/shRAP1 condition and Figure 3C for the +Dox conditions for number of fusions. Given the subtlety of the phenotype, a rescue experiment is crucial.
- 4) Figure 1C: there is concern over the small effect noted here (from ~9% to ~12%) as evidence for a primary function of Rap1 to protect telomeres in senescing cells. Perhaps a more efficient shRNA or another mode of Rap1 depletion could provide a larger and more convincing phenotype.

Figure 2

- 1) Figure 2A-F: All experiments here were performed only a single time and, thus, lack any statistical analysis. Thus, I cannot confidently be convinced of any of the effects here as reproducible. This same critique applies to Figure 3C and Figure 4A-C.
- 2) Figure 2C: A Western blot needs to be provided to show the level of knockdown for LIG3 and LIG4. Although mRNA levels are provided in Supplemental Figure 3E, LIG3 and LIG4 are very stable and resistant to knockdown. Thus, quantifying protein levels is necessary to draw conclusions for these experiments.
- 3) Figure 2C: shLIG3 and shLIG4 alone without Rap1 knockdown are required as controls for these experiments to ensure their knockdown does not affect fusion frequency even with Rap1 expressed fully. This same critique applies to Figure 3C.

Figure 3

- 1) Figure 3A-E: The use of clonal cell lines that have been genetically deleted of Rap1 (Rap1 Δ/Δ) would greatly strengthen the observations here.
- 2) Figure 3B: The amount of shortening with BIBR1532 here is not substantial. The effects seen here could be greater if cells are kept on inhibitor longer and telomeres become shorter.
- 3) Figure 3E. The increase in the percentage of end-to-end fusions is very minimal (~0.5% to ~1.5%). Longer treatment with telomerase inhibitor to allow more telomere shortening prior to harvesting metaphases spreads will provide a larger and more biologically relevant effect. Experiments here would benefit from a clean genetic experiment using knockout cell lines.

Figure 4

- 1) Figure 4: To study the effect of Rap1 on growth in cells that bypass senescence by knockdown of p21, the experimenters should perform their lentiviral infections at an earlier timepoint rather than after the cells have been senescent for multiple weeks. I would recommend adding virus at about 40 days into culture, one timepoint before the cells have entered senescence. In addition, the cells should be cultured longer after lentiviral infection (at least to 100 days in culture) to determine their fate. Do they crash out of culture and die? Do they just stay senescent permanently even with p21 knockdown? An assay for apoptosis versus senescence would be useful for this endeavor.

Referee #3:

In this manuscript Lototska and colleagues have revisited the long-standing debate surrounding the role of RAP1 at telomeres. While a direct function in protecting telomeres from end fusions is well established in budding and fission yeast, the involvement in end protection from fusions in mammalian cells has remained controversial. The current study presents intriguing new results suggesting that Rap1 plays a role in the protection of critically short telomeres from classical NHEJ specifically in senescent cells. The paper is well written and the results will be of substantial interest to the telomere and genome stability communities. My main concern relates to the preliminary nature of the analysis of fusions. The differences in the incidence of fusions are subtle compared to e.g. Trf2 deletion and the fusion PCR assay is notoriously noisy. The study would therefore be

substantially improved if each of the graphs in 2a, c, e, 3c, SF2d and SF3c) represented the average of biological triplicates (DNA isolated from three culture dishes subject to the fusion PCR) and error bars were included.

It is unclear to me why the authors conclude that "human RAP1 is required to protect critically short telomeres from classical NHEJ-mediated fusions". While the data supporting fusions and requirement for ligase 4 is clear (but see comment on replicates and error bars), it is less clear that the fusions are indeed occurring between critically short telomeres. Have the authors tried to sequence product of the fusion PCR? Most of the bands appear quite large compared to earlier studies examining critically short telomeres (e.g. Capper et al 2007).

The authors propose the interesting idea "that senescent cells that return to growth upon RAP1 depletion, rapidly succumbed to an excess of telomere and chromosome instability; for instance, the accumulation of chromosome fusions upon RAP1 knockdown triggers mitotic catastrophe,"

This can easily be tested in their system and including such data would significantly strengthen the model.

Other points:

It is unclear to this reviewer how the data in Supplementary Figure 1b was generated. Were the telomere signals from each IP divided by the input signal for Alu or by the barely visible Alu signal (single RAP1/TRF2 does not bind Alu sequences) for each IP?

An important piece of information that should be included in figure 1 is the average count of 53BP1 foci in each sample. Do 53BP1 foci (not TIFs) increase in senescent cells compared to presenescent cells independent of the RAP1 knock down? If 53BP1 staining is increased (as it looks in F1c, but is difficult to judge from a single cell), then the incidence of random 53BP1 / telo colocalization will be higher. If 53BP1 foci are difficult to count due to high density and overlap, quantification of 53BP1 positive volume per cell (or area in projections or single planes) can be used to compare samples

Considering that RAP1 is recruited to telomeres exclusively by TRF2, the 2-fold reduction of RAP1 versus 4 fold for TRF2 demands an explanation. Do the authors believe that this is measurement error or that RAP1 binding sites on TRF2 are not saturated in presenescent cells?

Page 8, line 4: "one-week post infection, most of the shp21CIP1 transduced cellshave lost their senescence associated β galactosidase (SA- β -gal) staining"; beta-gal staining is only shown for 15 days and is still at 41% at that time. It is unclear what is shown in Figure 4c lower panel. Is this an overlay with DAPI? The image does not help support the percentage of EdU incorporation. Each image should also contain a scale bar.

The reference 4 is cited as supporting a role of hRAP1 in protection against NHEJ in vitro. Work in this paper uses a cell-based system in which hRAP1 is recruited to telomeres independent of TRF2 in HeLa cells.

Page 2, line 5: "gene can be knocked out through"

1st Revision - authors' response

26 November 2019

Referee #1:

This is a very interesting paper shedding new light on the function of the TRF2 co-factor RAP1 and on telomere protection in human senescent cells, two key issues of telomere and cancer biology.

The authors convincingly show that RAP1 loss leads to a significant increase in 53BP1- LIG4-dependent telomere fusions specifically in senescent cells, not in dividing primary cells. This response is remarkably restricted to NHEJ since RAP1 loss does not result in other telomere defects

such as telomere fragility or telomere loss. Finally, RAP1 loss compromises cell viability when senescent cells are allowed to return to growth, a possible consequence of the increased frequency of chromosome fusions.

These results have important implications in the field of genome stability; hence, the paper will be read with interest by a wide audience.

We greatly appreciate your enthusiasm on our work.

Minor points:

1. The claim that RAP1 loss has no consequences in non-senescent dividing cells may be an overstatement. All the assays used here have a limited sensitivity and a significant background level. Caution regarding this point is necessary.

We agree with the reviewer that all assays have detection limitations. Of course, we cannot rule out that RAP1 has a protective role in non-senescent cells. For instance, we showed in HeLa cells that RAP1 depletion caused telomere fusion but when telomerase was chemically inhibited (Fig 4). Our results suggest that RAP1 is involved in the protection of critically short telomeres, thus, we propose in the final paragraph of this manuscript the following “One possibility is that RAP1 guarantees the stability of critically short telomeres that may occur on stochastically basis, during cell division”.

2. Page 7, 2nd paragraph: 53BP1 also promotes NHEJ through the Rif1 pathway.

This is a good remark, we had change the paragraph as follow : “Since 53BP1 promotes NHEJ by increasing chromatin mobility [27, 28] and/or by preventing end resection thanks to its interaction with RIF1 and the shieldin complex [29, 30], it is tempting to propose that the anti-fusion properties of RAP1 rely on its ability to prevent 53BP1 binding to telomeres.
”.

Referee #2

In this manuscript entitled "Human Rap1 specifically protects telomeres of senescent cells from DNA damage," Lototska et al. aim to establish the telomere-binding protein Rap1 as an essential factor for the protection of short telomeres in cells undergoing replicative senescence. The authors argue that Rap1 maintains telomere integrity of senescing cells by inhibiting recruitment of the DNA damage factor 53BP1 to telomeres and preventing telomere-telomere fusions via classical nonhomologous end joining in a LIG4-dependent manner. The authors argue that Rap1 protects short telomeres in senescing MRC-5 primary cells lacking telomerase expression and in Hela cells in which telomerase is inhibited chemically by BIBR1532. Lastly, the authors claim that Rap1 is necessary for senescent MRC-5 cells to re-enter the cell cycle and begin proliferation upon inhibition of the checkpoint protein p21.

Unfortunately, in its presented form, the data does not support the claim that Rap1 has a protective role at telomeres in aging cells. Many of the data presented is based on a single experiment and lacks appropriate statistical measures in order to interpret the results and make confident conclusions. In addition, several experiments lack fundamental (and critical) controls. The whole manuscript revolves on the use of a single shRNA. Minimally, the authors should use multiple RAP1 shRNAs to control for off-target effects. The key conclusions need to be strengthened by the use of clonal cell lines that have been genetically deleted of Rap1 (Rap1 Δ/Δ) and have been described in the literature. Overall, due to major concerns with the experimental data such as statistical rigor, lack of vital controls, and overinterpretation of minor phenotypic differences, this study is not suitable for publication. Specific concerns are as follows:

We apologise if the data were presented in a way that it was not clear to follow. We agree with the reviewer that more replicates and controls have to be added to strengthen our conclusions.

In the revised manuscript, we have repeated most of the experiments requested, as well as added new ones to improve this work and make it easier to follow (see specific details below).

Concerns:

Figure 1

1) Figure 1A: Previously, it has been established that TRF2 is necessary to maintain Rap1 protein and that loss of TRF2 leads to a loss of Rap1 (Celli et al., NCB). The authors report that TRF2 levels drop precipitously at late PD's; however, Rap1 levels remain mostly unchanged. A comment for this discrepancy is requested from the authors.

Indeed, it has been shown by Celli and de Lange (2005) that absence of TRF2 leads to RAP1 loss, however that study was performed in mouse embryonic fibroblast. For instance, in human cells, Takai et al., JBC (2010) have found that depletion of TRF2 using shRNAs lead to a mild decrease of RAP1 protein when assess by western blotting using whole cell lysates, as is the case in our work. In addition, and similar to our work, Swanson et al, (2016) had shown that during senescence the levels of TRF2 decrease but RAP1 remains stable and again only when the nuclear fraction is separated the effect is observed.

We had now added this reference in the text as follow: “As expected from replicative senescent cells and ageing animal models, the levels of TRF2 greatly decreased (around 80%) in senescent cells [13-15]. However, the levels of RAP1 barely changed with a mild decrease of only 15% as seen previously [16, 17].

2) Figure 1C: only one shRNA is used against Rap1. A second independent shRNA is necessary to control for potential off-target effects. This same critique applies to Figures 2 and 4.

To rule out potential off-target effects, we had now used siRNAs. In the case of Fig. 1C (TIF assay) we used an siRNA against RAP1 by performing two consecutive transfections of 3 days each to get close the time we inhibited RAP1 by shRNAs (6 days for siRNA and 10 days for shRNA). In addition, we had also infected senescent cells with the full-length RAP1 in senescent MRC-5 transduced with an shRAP1 vector. As you can see in the revised Fig 1C, we do rescue the increase in TIFs in these cells.

Importantly, we found similar number of TIFs when we used an siRNA against RAP1 as compared to shRAP1 conditions.

In the same way, we have now used siRNAs in the fusion PCR assay against RAP1, LIG3 and LIG4 and again we found similar results as compared to shRNAs. This new data has been added in the main figures, as you can see below (new Fig. 2C) :

3) Figure 1C: a rescue experiment should be performed by, expressing a Rap1 cDNA that is resistant to shRNA and showing a subsequent decrease of 53BP1 telomere foci to shScrb1 levels. This same critique applies to Figure 2C in the shLIG3/shRAP1 condition and Figure 3C for the +Dox conditions for number of fusions. Given the subtlety of the phenotype, a rescue experiment is crucial.

We do agree that the rescue experiment is important to confirm our model. Hence, for the mentioned experiments alongside downregulation of RAP1 we have now overexpressed RAP1 in senescent MRC-5. As you can see in the point above, we had included that data in the new Fig 1C (TIF assay) and Fig 2C (fusion assay). We had also shown in the first submitted version of this manuscript that ΔMyb truncation of RAP1 is also able to rescue fusion events but not the C-terminal ΔRCT mutant form. All these experiments had been performed in triplicates and error bars had been included in the graphs.

4) Figure 1C: there is concern over the small effect noted here (from ~9% to ~12%) as evidence for a primary function of Rap1 to protect telomeres in senescing cells. Perhaps a more efficient shRNA or another mode of Rap1 depletion could provide a larger and more convincing phenotype.

Indeed, the effect is not enormous but is reproducible and specific. i) We found statistically significant increase in cells treated with an shRNA against RAP1, ii) similar results were obtained after transient transfections with an siRNA against RAP1, iii) telomere damages is rescued by the expression of RAP1.

The downregulation obtained with either shRAP1 or siRAP1 in senescent cultures was at least 90% in all cases. We now provide the protein quantifications in all western blots.

Figure 2

1) Figure 2A-F: All experiments here were performed only a single time and, thus, lack any statistical analysis. Thus, I cannot confidently be convinced of any of the effects here as reproducible. This same critique applies to Figure 3C and Figure 4A-C.

Key experiments (such as the depletion of RAP1 in senescent cells or –RAP1 depletion by CRISPR in HeLa cells fusion PCR assay) were performed at least twice in the original version of this manuscript but presented separately (mainly in supplementary figures). We understand that it was confusing to follow therefore we had now combined all the results in a single graph and performed further replicates for the conditions where only one replicate was performed originally (as shown above).

All the fusion PCR assay Southern blot membranes can be found in the Source Data file.

2) Figure 2C: A Western blot needs to be provided to show the level of knockdown for LIG3 and LIG4. Although mRNA levels are provided in Supplemental Figure 3E, LIG3 and LIG4 are very stable and resistant to knockdown. Thus, quantifying protein levels is necessary to draw conclusions for these experiments

We have performed the western blot with anti-Ligase III and anti-Ligase IV antibodies. The results are included in the new figures EV2 and EV3.

3) Figure 2C: shLIG3 and shLIG4 alone without Rap1 knockdown are required as controls for these experiments to ensure their knockdown does not affect fusion frequency even with Rap1 expressed fully. This same critique applies to Figure 3C.

We added the suggested conditions to the graph. As you can appreciate (Fig. 2C-shown above and new Fig. 4C -below), downregulation of LIG3 or LIG4 alone does not increase the number of fusions. As expected, this number increases upon RAP1 downregulation in shLIG3/siLIG3 treated cells, but not in shLIG4/siLIG4 (Fig 4C).

Figure 3

1) Figure 3A-E: The use of clonal cell lines that have been genetically deleted of Rap1 (Rap1 Δ/Δ) would greatly strengthen the observations here.

We agree that using clonal cell lines depleted of RAP1 could strengthen the observations presented in this manuscript. To our knowledge, the cells that have been described in the literature, are either single clonal lines (Kabir 2014) or the doxycycline-inducible CRISPR knockout system of human RAP1 described in Kim et al (2017). We decided to take advantage of the latter system and so, all the data presented in Fig 4 are derived from clonal CRISPR HeLa cells. In summary, the results obtained in primary MRC-5 cells (shRNA and siRNA) together with CRISPR depletion of RAP1 in HeLa cells had been consistent allowing us to draw the conclusion presented in the manuscript.

2) Figure 3B: The amount of shortening with BIBR1532 here is not substantial. The effects seen here could be greater if cells are kept on inhibitor longer and telomeres become shorter.

To answer this concern, we have treated cells with the BIBR1532 for 50 days. Unfortunately, were unable to detect further telomere shortening compared to the 25 days already presented in the manuscript. We don't know whether that is the maximum shortening that BIBR1532 could generated in our experimental conditions (see figure bellow).

However, we had repeated the PCR fusion assay with 25-day BIBR1532 treatment to have more replicates and we had added further controls (-DOX shLIG3, -DOX shLIG4). Combined, these results again showed the protective role of RAP1 only when telomerase is inhibited and telomeres shorten, even if it is not substantial. Error bars and statistical significance has been added (as seen in Fig 4C).

3) Figure 3E. The increase in the percentage of end-to-end fusions is very minimal (~0.5% to ~1.5%). Longer treatment with telomerase inhibitor to allow more telomere shortening prior to harvesting metaphases spreads will provide a larger and more biologically relevant effect. Experiments here would benefit from a clean genetic experiment using knockout cell lines.

Please see the 2 previous comments above.

Figure 4

1) Figure 4: To study the effect of Rap1 on growth in cells that bypass senescence by knockdown of p21, the experimenters should perform their lentiviral infections at an earlier timepoint rather than after the cells have been senescent for multiple weeks. I would recommend adding virus at about 40

days into culture, one timepoint before the cells have entered senescence. In addition, the cells should be cultured longer after lentiviral infection (at least to 100 days in culture) to determine their fate. Do they crash out of culture and die? Do they just stay senescent permanently even with p21 knockdown? An assay for apoptosis versus senescence would be useful for this endeavor.

This is an interesting point. The reason we present the data at 15 days post infection was because we observed high levels of cell death and few cells remained attached to the surface of the plate. The cells cannot reach another population doubling. We have now investigated further the fate of the cells that return to growth upon p21 depletion. We had assayed apoptosis, as suggested, and found that after 15 days, 57% of the cells are apoptotic when RAP1 is depleted compared to 37% of control cells. Even more, most of the cells which are still attached to the surface of the culture plates show SA-B-gal staining suggesting they are back to senescence (92% compared to 43 % of shp21 cells). Potentially, shp21 cells could have been grown for longer time but the aim of this study was to see the effect of RAP1 in such cells, thus we stopped our control shp21 cells at the same time as the shp21+shRAP1 cultures.

We had added that new data in the revised figure 5E and 5F.

Referee #3:

In this manuscript Lototska and colleagues have revisited the long-standing debate surrounding the role of RAP1 at telomeres. While a direct function in protecting telomeres from end fusions is well established in budding and fission yeast, the involvement in end protection from fusions in mammalian cells has remained controversial. The current study presents intriguing new results suggesting that Rap1 plays a role in the protection of critically short telomeres from classical NHEJ specifically in senescent cells. The paper is well written and the results will be of substantial interest to the telomere and genome stability communities. My main concern relates to the preliminary nature of the analysis of fusions. The differences in the incidence of fusions are subtle compared to e.g. Trf2 deletion and the fusion PCR assay is notoriously noisy. The study would therefore be substantially improved if each of the graphs in 2a, c, e, 3c, SF2d and SF3c) represented the average of biological triplicates (DNA isolated from three culture dishes subject to the fusion PCR) and error bars were included.

We thank the reviewer for her/his very positive appraisal of the general interest of our findings.

We would like to apologize if some of the data were not presented clear enough or there were missing biological replicates. Therefore, for most of the experiments, we have added more replicates and combined already accomplished ones that were initially presented separately in

supplementary figures. As you can see, these changes apply to main figures, where the graphs represent the average of biological triplicates. Also, we performed statistical tests and added error bars.

It is unclear to me why the authors conclude that "human RAP1 is required to protect critically short telomeres from classical NHEJ-mediated fusions". While the data supporting fusions and requirement for ligase 4 is clear (but see comment on replicates and error bars), it is less clear that the fusions are indeed occurring between critically short telomeres. Have the authors tried to sequence product of the fusion PCR? Most of the bands appear quite large compared to earlier studies examining critically short telomeres (e.g. Capper et al 2007).

We thank the reviewer for raising this concern. We followed her/his advice on sequencing the products of fusion PCRs. To be able to fully sequence the whole length of the PCR products, we used long-read sequencing by Nanopore. We sequenced PCR products of senescent MRC-5 cells treated with either shControl or shRAP1. Interestingly, we found that the majority of the fusion events have very short telomere sequence, in fact for shControl 34% (16 out of 47) of the products sequenced had no telomere repeats at the fusion point. This compares to 10% in shRAP1 (9 out of 84 fusion events). The mean size of the telomeric array (including variant telomeric repeats) found at the junction of the fused chromosomes was 140 bp for shControl and 280 bp for shRAP1. For instance, when performing our PCR fusion assay, we also hybridized some membranes with a telomeric probe, but we were unable to detect a signal. Now, we know that the telomere array was too short to be detected and even more, in all PCR products we sequenced we found telomere variant repeats suggesting we are looking at the start of the telomere array as shown in the examples of the new Fig 3B. These new results strengthen our findings that RAP1 protects critically short telomere.

We have added a new figure (Fig 3) showing these results, distribution of the telomere array found at the junction point (below) and some sequencing examples. The data is already available in the Sequence Read Archive (SRA) repository through the following link: (<https://www.ncbi.nlm.nih.gov/sra/PRJNA577013>)

The authors propose the interesting idea "that senescent cells that return to growth upon RAP1 depletion, rapidly succumbed to an excess of telomere and chromosome instability; for instance, the accumulation of chromosome fusions upon RAP1 knockdown triggers mitotic catastrophe,"

This can easily be tested in their system and including such data would significantly strengthen the

model.

We had investigated further the fate of these cells by assaying for apoptosis. As mentioned above, we had found that around 37% of the cells that return to growth are apoptotic whilst this percentage increased to 57 % in cells with RAP1 inhibition, showing that apoptosis is one of the reasons why cells expressing an shRAP1 after they return to growth by inhibition of p21CIP die. We had changed the text as follow “In order to investigate this further, we measured the levels of apoptosis in our cultures (Fig 5E and 5F). Interestingly, RAP1 depletion in cells that return to growth caused significantly higher levels of apoptosis compared to control cells (Fig 5E and 5F), more likely by the excess of telomere and chromosome instability.”

This data has been added in the new Fig 5E.

Other points:

It is unclear to this reviewer how the data in Supplementary Figure 1b (Is it Sup Fig 1a?) was generated. Were the telomere signals from each IP divided by the input signal for Alu or by the barely visible Alu signal (single RAP1/TRF2 does not bind Alu sequences) for each IP?

We apologise for the way these data were presented. For the sake of clarity, we removed the graph of the normalization of Telo/Alu and we added a clearer description in the legend of Fig 1A.

An important piece of information that should be included in figure 1 is the average count of 53BP1 foci in each sample. Do 53BP1 foci (not TIFs) increase in senescent cells compared to presenescent cells independent of the RAP1 knock down? If 53BP1 staining is increased (as it looks in F1c, but is difficult to judge from a single cell), then the incidence of random 53BP1 / telo colocalization will be higher. If 53BP1 foci are difficult to count due to high density and overlap, quantification of 53BP1 positive volume per cell (or area in projections or single planes) can be used to compare samples

We have now counted the number of 53BP1 spots. As you can see in Fig. 1C (below), the average number of 53BP1 per nucleus increases with increasing population doublings. However, we did not observe a difference between shControl and shRAP1 condition in any of the time points analysed. Therefore, we can argue that the effect of RAP1-compromised senescent cells is telomere specific.

Considering that RAP1 is recruited to telomeres exclusively by TRF2, the 2-fold reduction of RAP1 versus 4 fold for TRF2 demands an explanation. Do the authors believe that this is measurement error or that RAP1 binding sites on TRF2 are not saturated in presenescent cells?

We thank the reviewer for her/his interesting comment. As the reviewer mentioned, it is possible that the saturation of TRF2 by RAP1 is different during senescence but also across the telomeric array. We can speculate that RAP1 preferentially binds to the TRF2 molecules present at the end of the telomeric array, thus during senescence, as the telomeres shorten, the decrease of TRF2 is more important.

We had added this remark in the main text as: “The density at telomeres of both TRF2 and RAP1 was reduced in senescent cells (PD 72) as compared to young cells (PD 30), but it was still detectable (Fig 1B). Interestingly, the density of telomere-bound RAP1 is reduced by only two-fold in senescent cells compared to a reduction of more than four-fold for TRF2. This apparent discrepancy could stem from differences in the binding of RAP1 to TRF2 across the telomeric array. It is possible that RAP1 binds preferentially the molecules of TRF2 found at the end of the telomeric array”.

Page 8, line 4: "one-week post infection, most of the shp21CIP1 transduced cellshave lost their senescence associated β galactosidase (SA- β -gal) staining"; beta-gal staining is only shown for 15 days and is still at 41% at that time. It is unclear what is shown in Figure 4c lower panel. Is this an overlay with DAPI? The image does not help support the ppercentage of EdU incorporation. Each image should also contain a scale bar.

We apologise for the wrong display of the data. We corrected the mistakes, more specifically, we now show the data for EdU and SA- β -gal each representing 2 time points: day 1 and day 15 post-infection with shControl, shp21 or shp21/shRAP1 lentiviral vectors. The EdU images represent the EdU staining (in magenta) together with an overlay with DAPI (blue). We also added scale bars and show the results of three independent replicates.

In addition, part of the data presented in Table 1 was combined with the data in the growth curve (Fig 5A) and together with new replicates we had added error bars in the growth curve of shp21CIP and shp21CIP + shRAP1 at 8 days and 15 days. By doing so we had eliminated Table 1 to make the results clearer to follow.

The reference 4 is cited as supporting a role of hRAP1 in protection against NHEJ *in vitro*. Work in this paper uses a cell-based system in which hRAP1 is recruited to telomeres independent of TRF2 in HeLa cells.

We thank for pointing this out. Indeed, the study in Sarthy et al., (2009) shows both *in vitro* and *in vivo* approaches. We had now added an introduction in to the manuscript and reference 4 is reference 9 in the revised manuscript. We had added that remark in the introduction as follow:

“In vitro, human RAP1 has been shown to protect against NHEJ in cooperation with TRF2 [8-10]. Furthermore, it was demonstrated *in vivo* that artificially targeting RAP1 to telomeres in a TRF2-independent manner can mediate NHEJ [9].”

Page 2, line 5: "gene can be knocked out through"

Corrected

2nd Editorial Decision

23 January 2020

Thank you for your patience while your revised manuscript was peer-reviewed at EMBO reports. We have now received the full set of reports as well as cross-comments; all pasted below.

As you will see, while referee 2 is more critical, both referees 1 and 3 support the publication of your revised study now. Please address all remaining concerns and send us a final manuscript file, along with a point-by-point response.

A few other changes are also required:

- please move the data availability section to the end of the materials and methods
- I attach to this email a related manuscript file with comments by our data editors. Please address all comments in the final manuscript file using the track changes option.
- please send us up to 5 keywords
- please check the abbreviation for JX Yu in the author contribution list
- Fig 2B+D need dividing lines or bigger spaces between the different gels. Fig 2D bottom right panel also has an extra 'splice'
- Fig EV2B needs dividing lines or spaces between the gels. It is also overcontrasted
- Fig EV3D is overcontrasted.
- many of the source data panels are also overcontrasted.

EMBO press papers are accompanied online by A) a short (1-2 sentences) summary of the findings and their significance, B) 2-3 bullet points highlighting key results and C) a synopsis image that is 550x200-400 pixels large (the height is variable). You can either show a model or key data in the synopsis image. Please note that text needs to be readable at the final size. Please send us this information along with the revised manuscript.

I look forward to seeing a final version of your manuscript as soon as possible. Please let me know if you have any questions.

REFEREE REPORTS

Referee #1:

This revision version fully addresses my requests. The proposed model that RAP1 protects critically short telomeres is convincing.

Referee #2:

The authors only partially addressed my technical concerns (see below). I still believe that the manuscript suffers from a major conceptual flaw. The authors argue that RAP1 specifically protects telomeres in senescence cells. Yet, they show that deletion of RAP1 in cancer cells, that do not senesce even in absence of telomerase, triggers telomere fusion. This is a fundamental inconsistency in the data. The authors must uncover this "enigmatic" characteristic that is present in senescent cells that renders telomeres more vulnerable to RAP1 loss.

1. The authors provide evidence from the literature that Rap1 levels are not dependent on TRF2 in all contexts, as was shown by shRNA in several cell lines in Takai et al., JBC (2010). They also reference Swanson et al. (2016) the data in the latter paper is uninterpretable as it is poorly controlled.
2. For all experiments that relied a single shRNA targeting Rap1, the authors now provide an additional siRNA. There is no Western in Figure 1 that would indicate the level of knockdown by this siRNA
3. To rule out off-target effect, the authors rescue shRNA depleted cells with a Rap1 cDNA. Nowhere do they indicate that the cDNA is resistant to the shRNA that targets exon 2. Why does the

shRNA vector not target the cDNA?

4. They now have performed replicates of fusion PCR experiments and statistical analysis in response to our concern having only done these experiments a single time without statistics. The evidence for Rap1 preventing fusions in their senescent cells is now convincing.
5. They now include the shLIG3 and shLIG4 alone conditions as requested, showing that the effects they see here are due to Rap1 knock-down in a LIG3/LIG4 knock-down background and not due to LIG3/LIG4 knock-down alone.
6. The authors are not able to get further shortening with BIBR when they extend to 50 days instead of 25 days. The rationale for this is not convincing.
7. The rationale for doing the p21 knock-down infection after cells have senesced rather than before senescence is not convincing.

Referee #3:

In this revised version of the manuscript the authors have addressed my main concerns regarding the reproducibility of their data. With three exceptions noted below the authors now include biological triplicates for the experiments as well a statistical analysis. Although the effects are modest, the authors make a compelling case in support of a role for Rap1 in the protection of very short telomeres in senescent cells.

Remaining concerns:

The Southern for BIBR1532 treatment for 50 days included in the comments to the reviewers shows convincing albeit modest telomere shortening. The Southern blot included in the manuscript showing treatment for 25 days shows a weaker signal for the BIBR plus lanes, but no convincing shortening. The apparent difference may be due to differences in loading. In conclusion, the 50 day timepoint results and analysis should be included in the manuscript.

Figure 2A shows a graph, the essential information regarding number of experiments and statistical test used to determine significance are missing. It is unclear whether the "Data information" statement at the end of the figure legend applies to panel A, C or both. Information on biological replicates and how many times the experiment was carried out are also missing for Figures EV3B and D.

Cross-comments by referee 1:

Reviewer #2 would like that the authors to uncover this "enigmatic" characteristic that is present in senescent cells that renders telomeres more vulnerable to RAP1 loss.

I think they did. This characteristic is the presence of critically short telomeres in senescent and cancer cells, telomeres too short to efficiently back-up RAP1 loss.

One simple interpretation: short telomeres bind less TRF2, therefore weakening the RAP1-independent protection pathways established by TRF2.

Cross-comments by referee 3:

In light of the revisions I would support publication of this work. This is a controversial (and emotionally surprisingly charged) area of telomere biology. Although the effects described in this manuscript are modest, in aggregate they make a case for a role of Rap1 in protection. Differences with previously published work can at least in part be explained by differences in the experimental

systems. I think this work will stimulate further investigation and should thus be made available to the scientific community.

2nd Revision - authors' response

26 January 2020

Referee #1:

This revision version fully addresses my requests. The proposed model that RAP1 protects critically short telomeres is convincing.

Referee #2:

The authors only partially addressed my technical concerns (see below). I still believe that the manuscript suffers from a major conceptual flaw. The authors argue that RAP1 specifically protects telomeres in senescence cells.

Yet, they show that deletion of RAP1 in cancer cells, that do not senesce even in absence of telomerase, triggers telomere fusion. This is a fundamental inconsistency in the data. The authors must uncover this "enigmatic" characteristic that is present in senescent cells that renders telomeres more vulnerable to RAP1 loss.

Our results show that RAP1 loss triggers telomere deprotection specifically in senescent cells as compared to young cells where RAP1 inhibition does not lead to telomere damages. Then we tested the hypothesis that this senescence effect was due to critical telomere shortening by using HeLa cells treated with a telomerase inhibitor to generate short telomeres. We found that RAP1 has telomere protective effects only upon telomere shortening, allowing us to conclude that short telomeres are specifically protected by RAP1. Therefore, there is no contradiction between the effect in senescent cells as compared to young ones and in HeLa cells upon critical telomere shortening. To avoid any confusion, we made clearer in the text of the revised version that the effect of Rap1 in senescent is specific as compared to corresponding young cells, as follows (page 6, line 9): “Together, these results show that the protective role of RAP1 is specific of senescent cells, as compared to dividing (young or pre-senescent) cells”.

In addition, we had added the following sentence in page 8, line 5: “However, this does not exclude that in other situations e.g. in cancer cells, critically short telomeres are protected by RAP1”.

Finally, to avoid any confusion, we have removed the word “specifically” from: Page1, line 5; Page 4, line 13 and Page 5, line 20.

1. The authors provide evidence from the literature that Rap1 levels are not dependent on TRF2 in all contexts, as was shown by shRNA in several cell lines in Takai et al., JBC (2010). They also reference Swanson et al. (2016) the data in the latter paper is uninterpretable as it is poorly controlled.

We thank the referee for considering our answer to his/her comment. We refer to Swanson et al study since it clearly shows by Western blotting that the levels of TRF2 substantially decrease in old cells but not RAP1. Those Western blots were done in a similar cellular background as us (human primary fibroblast).

2. For all experiments that relied a single shRNA targeting Rap1, the authors now provide an additional siRNA. There is no Western in Figure 1 that would indicate the level of knockdown by this siRNA

We agree with the reviewer. The siRAP1 experiment presented in Fig. 1C are MRC-5 senescent cells (PD72+4w, 3 biological replicates) and thus corresponds to the same experimental setting than Fig 2C, for which we already provided a WB (Fig. EV2D).

3. To rule our off-target effect, the authors rescue shRNA depleted cells with a Rap1 cDNA.

Nowhere do they indicate that the cDNA is resistant to the shRNA that targets exon 2. Why does the shRNA vector not target the cDNA?

The shRNA used in this study targets both the endogenous and RAP1 cDNA allowing us to avoid overexpression of the exogenous cDNA above the physiological level while rescuing the effect of the interfering RNA. The rescue to a nearly physiological level of RAP1 is presented in Fig. EV1.

4. They now have performed replicates of fusion PCR experiments and statistical analysis in response to our concern having only done these experiments a single time without statistics. The evidence for Rap1 preventing fusions in their senescent cells is now convincing.

We appreciate the positive appraisal on the role of RAP1 in preventing fusion events in senescent cells.

5. They now include the shLIG3 and shLIG4 alone conditions as requested, showing that the effects they see here are due to Rap1 knock-down in a LIG3/LIG4 knock-down background and not due to LIG3/LIG4 knock-down alone.

Indeed, the results show that the increase in fusions is due to RAP1 depletion.

6. The authors are not able to get further shortening with BIBR when they extend to 50 days instead of 25 days. The rationale for this is not convincing.

We agree with the referee that this was unexpected. Probably we are reaching some lower limit of telomere length to allow normal cell proliferation.

7. The rationale for doing the p21 knock-down infection after cells have senesced rather than before senescence is not convincing.

We do not understand this comment of the referee since the rationale of this experiment was to see the effect of the return to growth of senescent cells i.e. when cells are at their lowest telomere length. Thus, it is logic to inhibit p21 once the cells reached senescence and the return to growth is possible in replicative senescent cells upon p21-dependent checkpoint inhibition, as previously suggested in Beausejour et al EMBO J (2003).

Referee #3:

In this revised version of the manuscript the authors have addressed my main concerns regarding the reproducibility of their data. With three exceptions noted below the authors now include biological triplicates for the experiments as well a statistical analysis. Although the effects are modest, the authors make a compelling case in support of a role for Rap1 in the protection of very short telomeres in senescent cells.

Remaining concerns:

The Southern for BIBR1532 treatment for 50 days included in the comments to the reviewers shows convincing albeit modest telomere shortening. The Southern blot included in the manuscript showing treatment for 25 days shows a weaker signal for the BIBR plus lanes, but no convincing shortening. The apparent difference may be due to differences in loading. In conclusion, the 50 day timepoint results and analysis should be included in the manuscript.

We thank the reviewer for raising this concern. We have measured the length of telomeres of the conditions shown in Fig 4B (25-day treatment) and of the Southern blot presented in the previous answer to reviewers (50-day treatment). The extend of shortening generated by BIBR1532 treatment +DOX was greater after 25 days (shortening of 1.2 kb: peak telomere length of 3.6kb in -BIBR vs 2.4kb +BIBR) compared to 50 days (shortening of 0.7 kb: 3.4kb in -BIBR vs 2.7kb +BIBR) of treatment.

We don't think the effect in length is due to loading differences since the interstitial telomeric bands present in the Southern blot (Fig 4B) have similar intensities indicating equal loading. It is possible that this slight discrepancy between the 25 and 50 rate of shortening results from some lower limit of telomere length to allow normal cell proliferation leading to a counter-selection of cells exhibiting an excessive telomere shortening. In the revised version, we had added in Fig. 4B the measurement of telomere length, and the profile of the signal for +/-BIBR +DOX for 25 days in EV3A.

Figure 2A shows a graph, the essential information regarding number of experiments and statistical test used to determine significance are missing. It is unclear whether the "Data information" statement at the end of the figure legend applies to panel A, C or both. Information on biological replicates and how many times the experiment was carried out are also missing for Figures EV3B and D.

We had added the missing information in the figure legends.

Accepted

29 January 2020

I am very pleased to accept your manuscript for publication in the next available issue of EMBO reports. Thank you for your contribution to our journal.

Corresponding Author Name: MENDEZ-BERMUDEZ A, GILSON E, YE J

Journal Submitted to: EMBO REPORTS

Manuscript Number: EMBOR-2019-49076V1